# A Spinster-like Transporter at the Inner Membrane Complex is critical for *Toxoplasma gondii* cytokinesis, motility and invasion

Syrian G. Sanchez[1], Romuald Haase[1], David J. Dubois[1], Margaux Héritier[2,3¤],
Rachel Humann[1], Nicolas Hulo[4], Bohumil Maco[1], Isabel Meister[2], Leonardo Scapozza[2,3],
Oscar Vadas[1], Dominique Soldati-Favre[1]*

**1** Department of Microbiology and Molecular Medicine, Faculty of Medicine, University of Geneva, Geneva, Switzerland, **2** School of Pharmaceutical Sciences, University of Geneva, Geneva, Switzerland, **3** Institute of Pharmaceutical Sciences of Western Switzerland, University of Geneva, Geneva, Switzerland, **4** Institute of Genetics and Genomics in Geneva (iGE3), University of Geneva, Geneva, Switzerland

¤ Current address: Computational Structural Biology Unit, CNRS UMR 3528, Institut Pasteur, Université Paris Cité, Paris, France
* Dominique.Soldati-Favre@unige.ch

## Abstract

The Major Facilitator Superfamily (MFS) comprises a large and diverse group of membrane transport proteins involved in the translocation of metabolites across cellular membranes. The genome of *Toxoplasma gondii* encodes approximately 60 putative MFS transporters, yet the functions of most remain poorly characterized. Conserved across the superphylum Alveolata, the inner membrane complex (IMC) is a specialized peripheral membrane system essential for parasite replication, structural integrity, motility, and host cell invasion. Here, we identify *Toxoplasma gondii* Daughter Cell Transporter 1 (TgDCT1), a previously uncharacterized MFS transporter, as a critical regulator of daughter cell formation. TgDCT1 localizes predominantly to the daughter cell IMC and contains a predicted spinster-like MFS domain. Phylogenetic and structural analyses reveal that TgDCT1 is conserved across Alveolata, shares a canonical MFS fold with its *Plasmodium falciparum* orthologue, and exhibits striking structural similarity to the human sphingosine-1-phosphate (S1P) transporter SPNS2, suggesting an evolutionarily conserved role in lipid transport. Conditional depletion of TgDCT1 results in severe defects in cytokinesis, including disrupted IMC architecture, aberrant daughter cell morphology, and failure of plasma membrane abscission. Although TgDCT1-depleted parasites retain the capacity for microneme secretion and egress, they display profoundly impaired motility and host cell invasion, ultimately leading to arrest of the lytic cycle. Notably, pharmacological inhibition of the S1P transporter SPNS2 using the compounds 11i and 33p phenocopies TgDCT1 depletion, impairing parasite morphogenesis, intracellular replication, and division synchrony. Furthermore, transgenic complementation demonstrates

**Data availability statement:** All relevant data are within the paper and its Supporting information files.

**Funding:** This work was funded by grant HUG RC09-18 from the Private Foundation of the Geneva University Hospitals (https://www.fondationhug.org/) and by grant TMAG-3_216166 from the Swiss National Science Foundation (https://www.snf.ch/en), both awarded to D.S.-F. The funders did not play any role in the study design, data collection and analysis, decision to publish, or preparation of the manuscript.

**Competing interests:** The authors have declared that no competing interests exist.

that the spinster-like domain of the *P. falciparum* DCT1 orthologue can functionally substitute for TgDCT1, indicating that these transporters likely recognize the same substrate. Together, these findings establish TgDCT1 as a central regulator of lipid homeostasis required for IMC maturation, endodyogeny, and parasite propagation in *Toxoplasma gondii* and likely other Apicomplexa.

## Author summary

The inner membrane complex (IMC) is a unique cellular structure crucial for the replication, shape, and motility in *Toxoplasma gondii*, which causes widespread infections in humans and animals. Our study focuses on TgDCT1, a previously uncharacterized transporter protein specifically localized to the IMC of daughter parasites during cell division. TgDCT1 belongs to the Major Facilitator Superfamily of membrane transporters and possesses features indicative of a role in lipid transport. Notably, this transporter is conserved across multiple Alveolata, underscoring its evolutionary significance beyond *Toxoplasma*. Using various knockdown systems, we demonstrate that TgDCT1 is essential for IMC homeostasis; its depletion results in severe morphological defects, including abnormal cytokinesis and compromised IMC integrity. Although parasites lacking TgDCT1 can still exit host cells, their motility and invasion capabilities are dramatically impaired, preventing successful completion of the lytic cycle. These findings establish TgDCT1 as a key player in *T. gondii* development, suggesting that its transported substrate is vital for maintaining IMC structure and parasite viability. Targeting such critical transport mechanisms could open new avenues for therapeutic intervention against apicomplexan parasites.

## Introduction

The phylum of Apicomplexa comprises a diverse group of obligate intracellular eukaryotic parasites responsible for severe human and veterinary diseases. Notable members include *Plasmodium* spp., *Cryptosporidium* spp., and *Toxoplasma gondii*, the causative agents of malaria, cryptosporidiosis, and toxoplasmosis, respectively [1–3]. *T. gondii* is a ubiquitous parasite capable of infecting virtually all warm-blooded vertebrates, including humans, with an estimated one-third of the global population chronically infected [4]. While infection is typically asymptomatic in immunocompetent individuals, it can cause severe disease in immunocompromised patients or following congenital transmission. The pathogenicity of *T. gondii* is primarily driven by the rapidly dividing tachyzoite stage, which undergoes a specialized form of asexual replication known as endodyogeny. This process involves the synchronous intracellular formation of two daughter cells (DCs) within a mature mother cell, enabling efficient parasite proliferation within the host [5]. During each division cycle, secretory organelles such as micronemes, rhoptries, and dense granules are synthesized de novo

to support invasion and subversion of host cell functions [6], while other cellular structures are duplicated and inherited in a highly coordinated manner [7]. This precise orchestration underlies the parasite's ability to repeatedly invade, replicate, and egress from host cells, processes that are central to the pathology of toxoplasmosis [8].

Apicomplexans, Colpodellida, Colponemida, Ciliates, Dinoflagellates, and Perkinsozoa form the Alveolata superphylum [9] and share a defining feature: submembrane flattened vesicles called alveoli [10–12]. In apicomplexans, these form the inner membrane complex (IMC), crucial for the parasite's shape, motility, and replication. The IMC is linked to an alveolin network of intermediate filament–like proteins, characterized in *Plasmodium* and *Toxoplasma* [13]. In *T. gondii*, 22 sub-pellicular microtubules (SPMTs) anchor beneath the IMC, originating from the apical polar ring to maintain cell polarity and rigidity [14–16]. Together, the IMC and the plasma membrane (PM) form the parasite pellicle, a structure where the membranes are separated by a narrow pellicular space. This compartment contains key components required for gliding motility, including the myosin motor A and gliding-associated proteins (GAPs), which are embedded in the alveoli and serve as molecular linkers between the IMC and the PM [17]. The pellicular space is a crucial compartment where the apico-basal flux of filamentous actin takes place to generate motility [17]. The IMC extends along the parasite and is divided into apical, medial, and basal regions with distinct protein subsets [18]. While the apical cap forms a continuous layer, the medial and basal regions consist of IMC plates joined by transverse and longitudinal sutures composed of specific proteins [19–21]. This extensive coverage of the parasite's inner surface limits vesicle exchange between the plasma membrane and the cytoplasm. To overcome this barrier and allow vesicle trafficking, specialized IMC structures are required. The apical annuli are ring-shaped structures between the apical cap and central IMC plates and implicated in dense granule exocytosis [22–25]. Another structure, the micropore, forms a PM invagination at IMC intersections. Supported by a protein ring, it facilitates endocytosis by enabling vesicle uptake into the cytoplasm [26,27].

The DC formation is a highly regulated process, relying on transcriptional control, temporal recruitment of specific proteins, and post-translational regulatory mechanisms [28]. During endodyogeny, centrosome duplication is followed by DC IMC assembly and cortical microtubule formation [29–34]. Most DC IMC proteins are synthesized de novo, though some maternal components are recycled [35]. Certain proteins are uniquely expressed in forming DCs and vanish upon maturation, indicating stage-specific roles [21,36–42]. Their functional importance is evident, as depletion often disrupts IMC assembly, morphology, and viability. IMC biogenesis follows a "just-in-time" model, with proteins recruited in a tightly regulated temporal sequence [43]. For instance, ISP1 was long considered the earliest DC marker [18], but newer studies have identified even earlier markers [37–41,44]. Some proteins localize exclusively to the maternal or daughter IMC, while others (like AC9, ISP1, GAP50, and IMC1) are sequentially recruited and retained in both [44–47], reflecting IMC specialization throughout the cell cycle. IMC formation is also coordinated with organelle biogenesis. While secretory organelles form de novo, others, such as the endoplasmic reticulum, Golgi, mitochondrion and apicoplast, divide and partition between daughters [7]. The apicoplast closely associates with budding IMCs during segregation [40,48]. Recent studies implicate IMC10 and myosin motor A in maintaining intracellular morphology and mitochondrial inheritance [49,50], underscoring the IMC's role as a structural scaffold for organelle partitioning during replication.

Although the protein compositions of daughter and maternal compartments have been extensively studied, many IMC proteins remain uncharacterized. Furthermore, despite the importance of this organelle, its exact metabolite composition, particularly regarding lipids, remains poorly understood and largely unexplored. Only a few studies have directly or indirectly investigated the lipid composition and distribution within the IMC, and the mechanisms governing lipid transport and homeostasis in this structure remain largely unknown [51–55].

The Major Facilitator Superfamily (MFS) is one of the largest families of secondary transporters, mediating the movement of diverse substrates (including ions, metabolites, and lipids) across membranes in all domains of life [56,57]. MFS transporters typically have 12 transmembrane α-helices forming a membrane-spanning pore and are classified by substrate specificity, transport mode (uniport, symport, antiport), and evolutionary lineage [58–61]. Their central roles in cellular physiology make them key targets in biomedical research. Among MFS subfamilies, the Spinster family represents

a distinct subgroup, characterized by 12-transmembrane-domain transporters specialized in sphingolipid transport in humans [62]. First identified in Drosophila for their role in programmed cell death and degradation of neuronal cells, they were later associated with lysosomal homeostasis and autophagy [63–65]. In vertebrates, Spinster proteins are best known through SPNS2, the major exporter of sphingosine-1-phosphate (S1P), regulating immune cell trafficking, vascular signaling, heart development, and tumor metastasis, among other physiological processes [66–68].

Importantly, several small-molecule inhibitors of SPNS2 have recently been developed and tested for their ability to block S1P export. These include SLF1081851 (16d) [69,70] and more potent aminobenzoxazole and phenyl-urea/benzox-azole derivatives such as SLB1122168 (33p) [71–73] and SLF80821178 (11i) [74], respectively. Together, these com-pounds constitute valuable chemical tools to investigate the biological and therapeutic consequences of SPNS2 inhibition.

In *T. gondii*, several MFS transporters localize to distinct compartments and contribute to functions such as metabolism, organelle maturation, and exocytosis [75–82]. In this study, we identified a putative transporter, Daughter Cell Transporter 1 (DCT1), uniquely localized to the alveoli of DCs. According to in silico prediction, DCT1 belongs to the Spinster-like subgroup of the MFS. Our data reveal that DCT1 is essential for IMC homeostasis, gliding motility, and invasion. Loss of DCT1 leads to pronounced IMC morphological defects and compromised its structural integrity, as shown by ultra-expansion microscopy (U-ExM) and transmission electron microscopy (TEM). While its substrate remains unknown, conserved orthologs with similar architecture and binding pocket residues are present across Alveolata, suggesting a conserved function. Lipidomic analyses of the DCT1 mutant revealed that conditional depletion of DCT1 disrupts the par-asite's lipid balance, highlighting its role in lipid homeostasis. Moreover, drugs originally targeting the SPNS2 transporter are effective against *Toxoplasma* and recapitulate the phenotypic defects observed in the DCT1 mutant. Overall, our findings position DCT1 as a key regulator of lipid and IMC homeostasis, opening new avenues to decipher the molecular mechanisms underlying parasite morphogenesis and lipid homeostasis.

## Results

### *T. gondii* DCT1 is an alveolate-conserved putative transporter located to the daughter inner membrane complex

In an *in silico* screen using ToxoDB (https://toxodb.org/toxo/app) for putative essential MFS transporters (Phenotype score < −3) [83] (S1 Table), we selected TGGT1_258700, without assigned localization based on Localization of Organelle Proteins by Isotope Tagging [84] and predicted to have 12 transmembrane domains by DeepTMHMM [85].

Submission of DCT1 amino acid sequence to InterPro (https://www.ebi.ac.uk/interpro/), identified a putative MFS Spinster-like domain (Fig 1A). Furthermore, BLAST searches against several reference strains representing diverse Alve-olata organisms revealed that the protein is highly conserved, suggesting a potentially common role within this supraphy-lum (S1A Fig for conservation table and S1 Table for BlastP scores and E-values).

To refine the identification of plausible DCT1 orthologs, we retrieved genes annotated with a predicted MFS domain (InterPro: IPR036259) from the VEupathDB database and used them to construct a phylogenetic tree. This search was restricted to a subset of Alveolata species (see the Materials and Methods section for the list of selected organisms, and S1 Table for the corresponding MFS domain-containing FASTA sequences). We then constructed a large-scale phylogenetic tree using the complete set of MFS-containing sequences (S1 Data). This approach allowed us to confirm the orthology of the candidate genes identified through sequence similarity searches within an evo-lutionary framework. Phylogenetic analysis confirmed the presence of TgDCT1 orthologs in both Apicomplexa and Colpodellida (Fig 1B). Orthologous proteins across the examined species consistently contained an MFS spinster domain of relatively conserved length (427–563 Amino acids). In contrast, N-terminal regions exhibited considerable sequence length divergence (55–810 amino acids) (Fig 1C). MFS domain alignments revealed conserved MFS trans-membrane domains in most DCT1 orthologs; however, structural deviations, such as deletions in transmembrane regions of *Hepatocystis piliocolobus*, indicate that some orthologs may have diverged functionally or could alterna-tively be mis-annotated (S2A Fig).

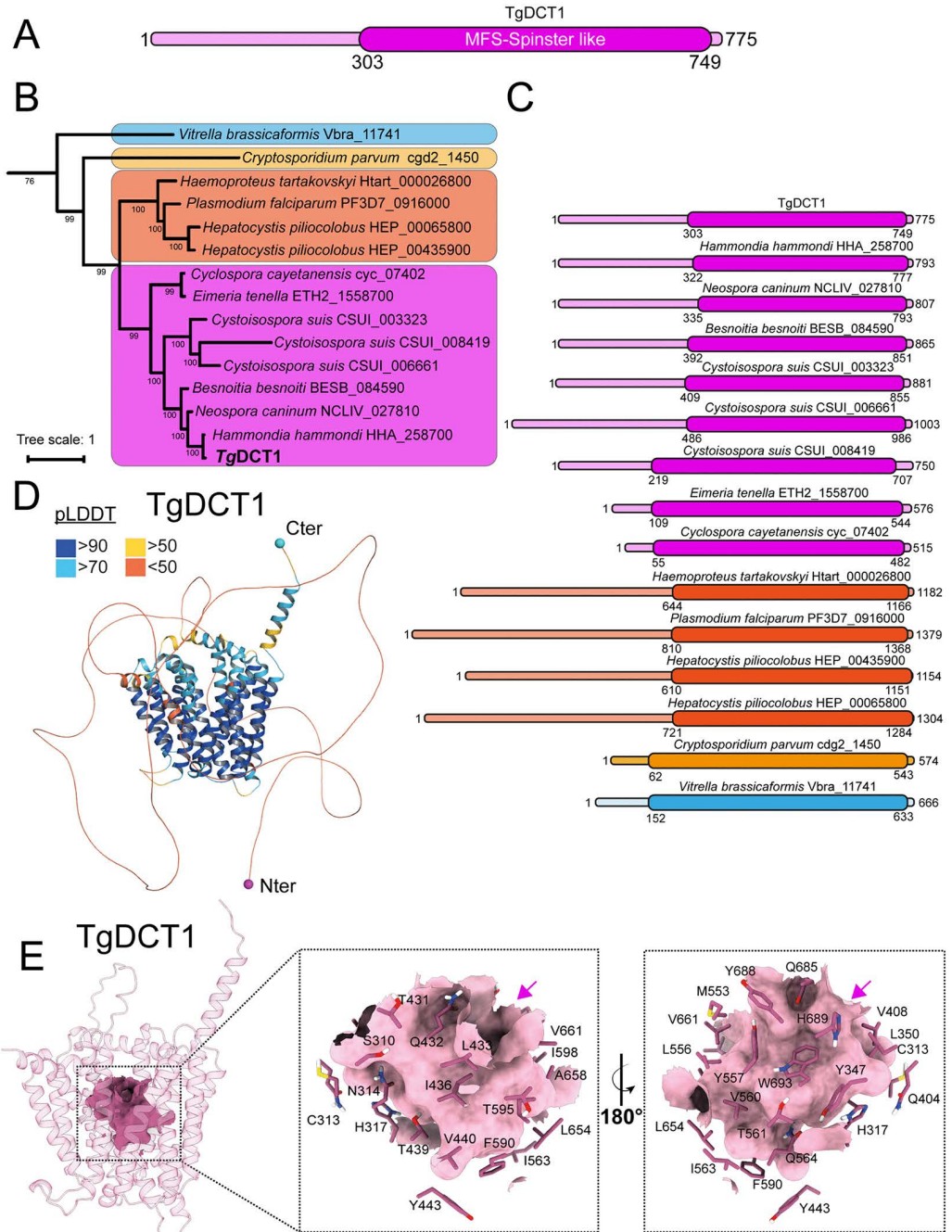

**Fig 1. DCT1 is a putative Spinster-like MFS transporter well conserved across the Alveolates. (A)** Schematic representation of the predicted domain organization of the protein. The numbers indicate amino acid positions corresponding to the full-length protein and highlight the start and end points of the predicted spinster-like domain. **(B)** Phylogenetic analysis of DCT1 orthologs within the Alveolata superphylum. The phylogenetic tree was constructed from a multiple sequence alignment of 897 putative MFS transporters identified across diverse Alveolata species. The tree was subsequently pruned to retain only the DCT1 orthologs, illustrating the evolutionary conservation of this transporter within the superphylum. The tree highlights taxa such as Colpodellida (blue), and various Apicomplexans, including Cryptosporidiidae (yellow), haemosporidians (orange), and coccidians (magenta). Bootstrap support values (based on 1000 replicates) are shown next to the branches, indicating the robustness of each clade. The scale bar represents the phylogenetic distance. Protein identifiers are indicated for each species. **(C)** Schematic representation of the predicted domain organization of DCT1 orthologs. The numbers indicate the amino acid positions of each full-length protein and mark the start and end points of the predicted spinster-like domain. **(D)** Predicted 3D structure of DCT1 (Alphafold3) colored according to pLDDT score. Magenta and cyan circles indicate the first

N-terminal and the last C-terminal residues of the predicted coding sequence, respectively. **(E)** Residues forming the putative binding pocket of DCT1. The binding pocket is highlighted in purple, while the majority of the transmembrane domains are shown in pink (top view). Insets show the residues surrounding the binding pocket that could potentially interact with a substrate (bottom view). Magenta arrows highlight the binding pocket in each inset.

To predict potential substrate conservation, we searched for residues that potentially contribute to transport specificity in the DCT1 structural model (Fig 1D) generated by AlphaFold3 (https://alphafoldserver.com/) [86]. Thirty-two residues were identified in the predicted binding pocket; all displayed high predicted local distance difference test (pLDDT) scores (Fig 1E), and most are conserved across orthologs, with only minor substitutions that generally preserve physicochemical properties (S2A Fig). Furthermore, the AlphaFold3 model of PF3D7_0916000 (PfDCT1) (S2B Fig) revealed an excellent Spinster domains structural alignment with the TgDCT1 model (S2C Fig), reinforcing the hypothesis that these two transporters recognize and transport the same metabolite.

To determine the DCT1 localization, we tagged the endogenous locus via homologous recombination, inserting a 2xTy epitope tag and a Dihydrofolate reductase (DHFR) cassette at the 3' end of the gene in the RH ΔKu80 (RH) strain. A clonal cell line was generated and validated by genomic Polymerase chain reaction (PCR) (S3A and S3B Fig). Expression of the C-terminally tagged DCT1 was further confirmed by western blot (WB), which revealed three distinct forms of DCT1-2Ty (Fig 2A). The predicted full-length DCT1-2Ty was detected at ~86 kDa, along with two additional forms at ~60 kDa and ~40 kDa. Notably, their relative abundance differed between intracellular and extracellular parasites: the higher molecular weight isoform, likely representing the unprocessed form, was more prominent in extracellular parasites, suggesting that DCT1 undergoes processing during intracellular development (S4A Fig). The localization of DCT1 was shown by immunofluorescence assays (IFA) to be restricted to the DC IMC and absent from the maternal IMC1 signal (Fig 2B). Additionally, the DCT1-Ty signal was observed in close proximity to the nucleus in extracellular parasites and did not show clear colocalization with the IMC (S4C Fig). During the early stages of DC formation, the DCT1-2Ty signal was already detectable at nascent buds, whereas ISP1 staining was not yet visible, indicating that DCT1 recruitment precedes the appearance of ISP1. This temporal pattern suggests that DCT1 may serve as a scaffolding nucleator during DC formation (Fig 2C). To determine whether DCT1 associates with the alveoli or the alveolin network, we conducted a solubility assay using Phosphate Buffer Saline (PBS), 1 M NaCl, 0.1 M $Na_2CO_3$, 1% Triton X-100, and 1% SDS. In this assay, soluble proteins are recovered in the supernatant, whereas insoluble proteins remain in the pellet. PBS extracts soluble proteins, 1 M NaCl disrupts electrostatic interactions, and 0.1 M $Na_2CO_3$ releases proteins weakly associated with membranes. Detergents such as Triton X-100 and SDS solubilize integral membrane proteins. Alveolar membrane proteins are soluble in both 1% Triton X-100 and 1% SDS, whereas alveolin network proteins are poorly soluble in 1% Triton X-100 but solubilized by 1% SDS. The results of this experiment showed that DCT1 was insoluble in PBS, NaCl, and $Na_2CO_3$, but was solubilized by Triton X-100 (TX-100) and SDS, indicating that it can be considered an integral alveoli membrane protein (Fig 2D).

## DCT1 is essential for IMC integrity and lytic cycle progression

To assess the role of DCT1, we generated an inducible knockdown line in the DiCre ΔKu80 ΔHXGPRT (DiCre) background by inserting a 3Ty-LoxP-HXGPRT-LoxP-U1 (3Ty-HX-U1) cassette at the 3′ end of the gene, creating the DCT1-Ty-U1 strain (S3C Fig). This modification enabled Rapamycin (Rapa)-inducible translational silencing of the gene via U1-mediated regulation [87]. Proper cassette integration and efficient excision under Rapa treatment were confirmed by diagnostic genomic PCR (S3D Fig). WB analysis showed complete DCT1 depletion 24h post-Rapa treatment and revealed three forms, consistent with the knock-in strain and their differential abundance in intra- and extracellular parasites (Figs 3A and S4A). IFA confirmed correct localization of DCT1 at the DC IMC and its loss following Rapa induction, which coincided with a severe morphological defect of the IMC (Fig 3B). Quantification revealed that ~87% (±5.5) of

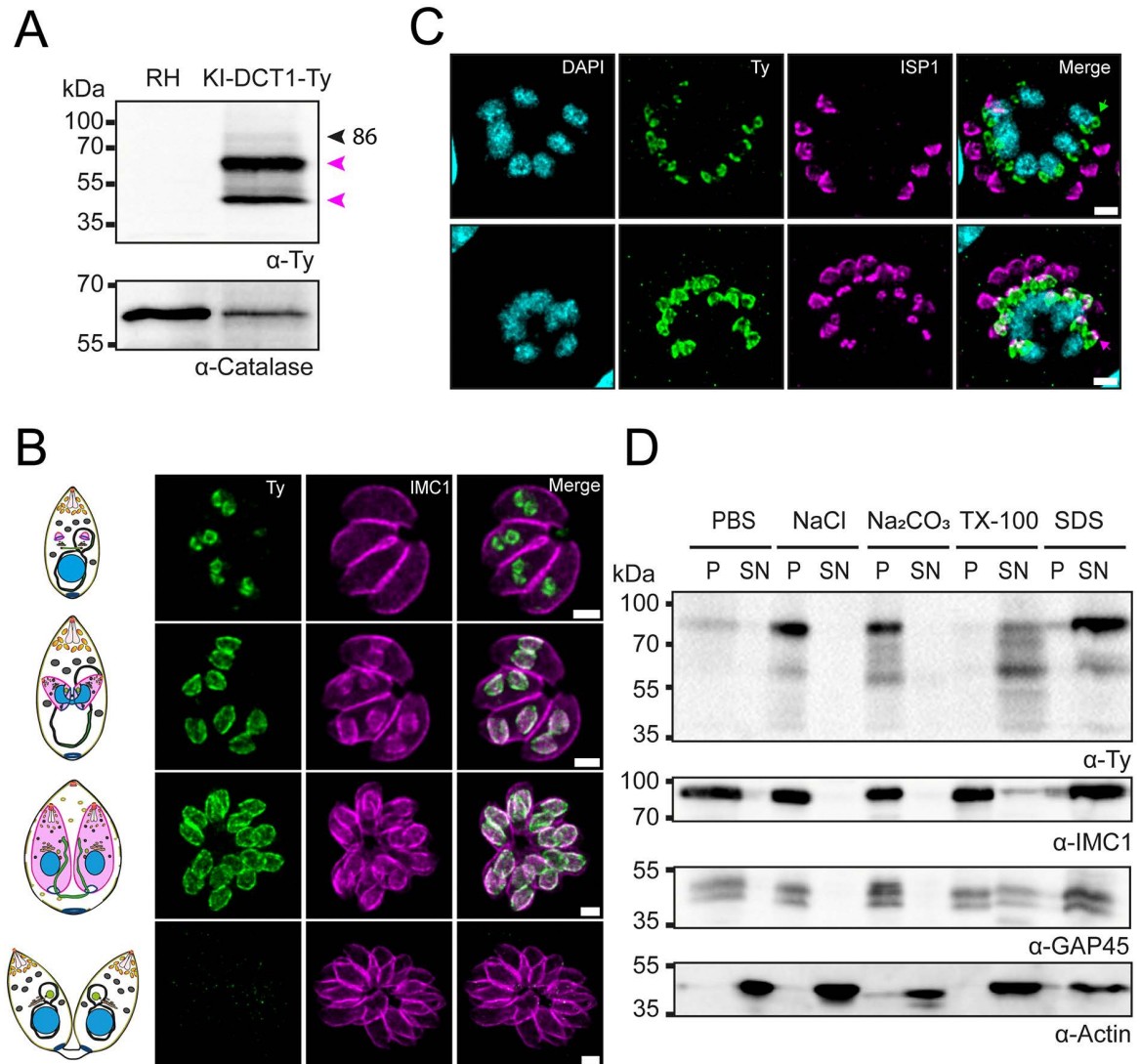

**Fig 2. DCT1 is a polytopic protein present only in the daughter IMC. (A)** Immunoblot analysis of parental RH and KI-DCT1-Ty intracellular parasite protein extracts 30 hours post-infection. DCT1-2Ty was detected using an anti-Ty antibody. Anti-Catalase was used as a loading control. The black arrow indicates the full-length form of the protein, while purple arrows denote additional forms of the tagged protein. **(B)** IFA on intracellular KI-DCT1-Ty parasites showing the localization of DCT1 throughout the endodyogeny process. DCT1 was detected using anti-Ty antibodies (green), while the IMC of both mother and daughter cells was labeled with anti-IMC1 antibodies (magenta). Scale bar: 1 μm. **(C)** Colocalization of DCT1 (anti-Ty; green) with ISP1 (anti-ISP1; magenta) indicates that DCT1 is an early marker of parasite endodyogeny. DNA was labeled with DAPI (cyan). Green arrows indicate nascent buds where DCT1-2Ty signal is already present before ISP1 becomes detectable, whereas magenta arrows highlight daughter buds where both DCT1-2Ty and ISP1 signals are visible. Scale bar: 2 μm. **(D)** Western blot analysis of DCT1-2Ty solubility in intracellular parasites highlights the protein's presence at the Alveoli. Solubility assay has been done using different buffers (PBS, 1 M NaCl, 0.1 M Na₂CO₃, 1% TX-100, and 1% SDS) in combination with parasite lysis by freeze-thaw cycles, resulting in a supernatant fraction (SN) and a pellet fraction (P). Anti-Ty was used to detect DCT1-2Ty. Anti-IMC1 and anti-GAP45 antibodies were used as controls to assess the solubility of the alveolin network and alveoli-associated proteins, respectively. Anti-Actin was used as a control for soluble proteins.

Rapa-treated vacuoles exhibited IMC defects, compared to ~9% (±3.5) in controls (Fig 3C). DCT1 absence also impaired lytic cycle progression, as shown by plaque assays: Rapa-treated parasites displayed an~88% (±3.3) reduction in plaque size compared to controls (Fig 3D and 3E).

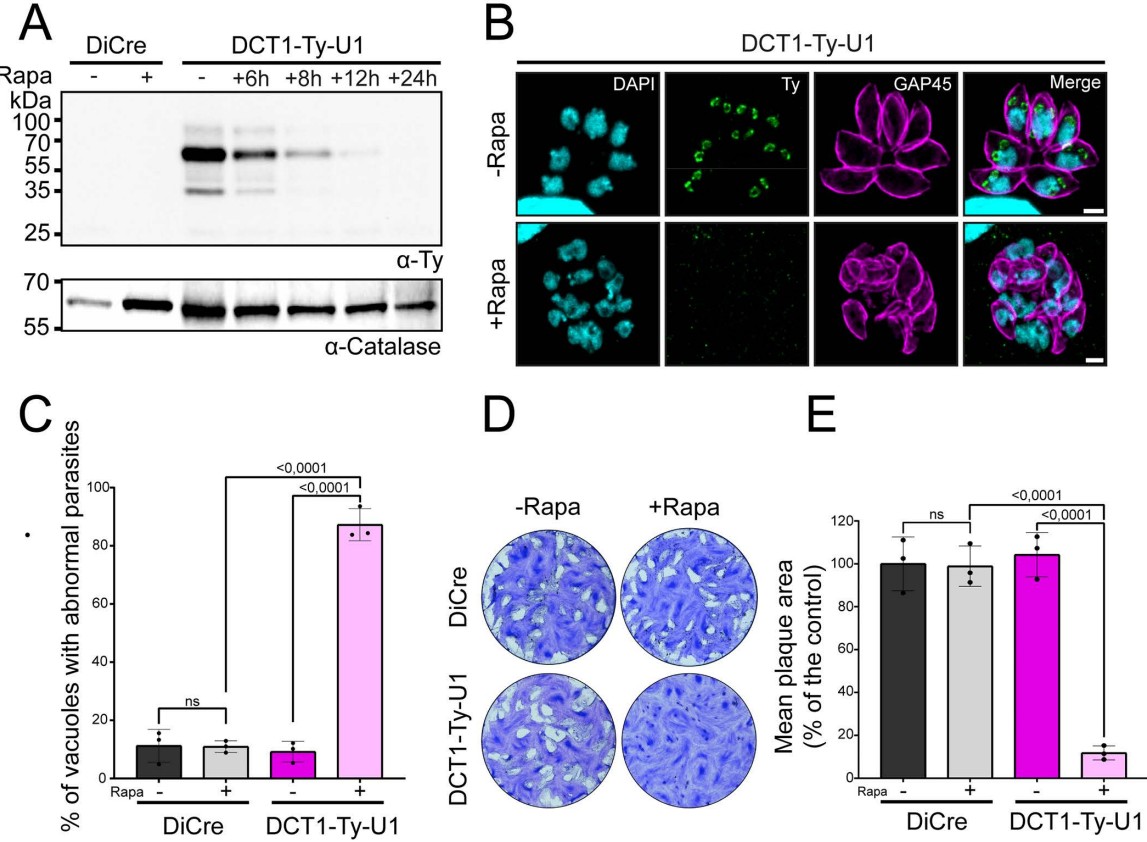

**Fig 3. Conditional depletion of DCT1 causes morphological defects and severely impacts the lytic cycle. (A)** Western Blot analysis of parental DiCre and DCT1-Ty-U1 intracellular parasites treated with or without Rapa, revealing the depletion kinetics of DCT1. DCT1 was detected using an anti-Ty antibody, and anti-Catalase was used as a loading control. **(B)** IFA on intracellular DCT1-Ty-U1 parasites ± Rapa treatment 30 hours post-infection, showing the disappearance of DCT1 and aberrant IMC morphology upon treatment. DCT1 was labeled with anti-Ty (green), the IMC with anti-GAP45 (magenta), and DNA with DAPI (cyan). Scale bar: 2 μm. **(C)** Graph reveals the percentage of vacuoles containing DCT1-Ty-U1 parasites with abnormal IMC morphology after 30 hours of Rapa treatment. Statistical differences between groups were assessed using one-way ANOVA followed by Tukey's multiple comparisons (mean ± SD; n = 3 biologically independent experiments). **(D)** Representative images of plaque assay from three independent experiments comparing the lytic cycle progression of parental DiCre and DCT1-Ty-U1 parasites ± Rapa. **(E)** Graph summarizing the results of plaque assays conducted on DiCre and DCT1-Ty-U1 parasites ± Rapa. The mean plaque area for each condition is normalized to the results obtained for the DiCre -Rapa condition. One-way ANOVA followed by Tukey's multiple comparison was used to test differences between groups (mean ± SD; n = 3 biologically independent experiments).

To assess DCT1 topology at the parasite alveoli and enable rapid protein depletion, we used the mini–Auxin Inducible Degron (mAID) system, which triggers degradation of cytoplasm-exposed mAID-tagged proteins upon indole-3-acetic acid (IAA) treatment [88]. A conditional DCT1 mutant in the TIR1 ΔKu80 ΔHXGPRT (TIR1) background was generated by replacing the endogenous 3' untranslated region (3' UTR) with a mAID-3HA-HXGPRT cassette, yielding the DCT1-mAID strain (S3E Fig). Integration was confirmed by diagnostic genomic PCR (S3F Fig). WB showed efficient DCT1-mAID-3HA depletion within one hour of IAA treatment (S4B Fig). IFA revealed a transient DCT1 accumulation near the nucleus likely corresponding to the Golgi apparatus. Additionally, WB showed higher levels of the ~95 kDa DCT1-mAID-3HA protein in intracellular parasites compared to other lines, suggesting that C-terminal tagging with mAID may interfere with protein processing, potentially leading to Golgi accumulation (S4A and S4B Fig). Similarly to the Ty-tagged cell lines, in untreated parasites, the DCT1-mAID-3HA signal disappeared after completion of mother cell formation, indicating a tightly regulated

expression pattern during the cell cycle (S4D Fig). Although DCT1-mAID was not detected in daughter cells, we speculate that a small amount of protein is present at the growing alveoli during daughter cell formation, becoming barely visible when the fluorescence signal was amplified. (S4E Fig). Consistent with its localization, the solubility assay performed on the DCT1-mAID cell line revealed that DCT1-mAID-3HA is an integral membrane protein, likely associated with the IMC alveoli, as it was fully solubilized by both Triton X-100 and SDS (S4F Fig).

Upon IAA treatment, DCT1-mAID parasites exhibited a complete loss of the protein signal after 30 hours, resulting in several IMC defects similar to those observed in the DCT1-Ty-U1 line, with ~89% (±5.5) of vacuoles affected versus ~9% (±2.6) in controls (Fig 4A and 4B).

Collectively, these results confirm efficient auxin-induced degradation of DCT1, indicating that the C-terminus domain of this protein is exposed to the cytoplasm. Moreover, DCT1-mAID-3HA exhibits dynamic cell-cycle–dependent localization while likely remaining stably associated with DC IMC membranes, despite predominant mislocalization of the bulk protein signal. Given the efficient regulation and lack of significant morphological defects in the absence of IAA, we used the DCT1-mAID strain to further investigate DCT1 function.

## Trans-genera complementation with the PfDCT1 Spinster-like domain complement DCT1-depleted parasites

To confirm that observed phenotypes were due to DCT1 loss rather than off-target effects, we generated a complemented line (DCT1-mAID/cDCT1-Ty) by integrating a cDNA copy of TgDCT1 fused to a 3Ty tag under the *T. gondii Tubulin1* promoter into the *UPRT* locus (S5A Fig). Correct integration was validated by genomic PCR and WB (S5B and S5C Fig). IFAs showed that IAA treatment efficiently depleted the endogenous DCT1-mAID-3HA while cDCT1-3Ty remained stably expressed and correctly localized. After 30 hours in presence of IAA, IMC morphology appeared normal, indicating that the expression of cDCT1-3Ty had rescued the mutant phenotype (Fig 4C). Plaque assays further confirmed the critical role of DCT1 with IAA treatment, leading to a reduction of plaque size by ~87% (±1.5) in DCT1-mAID parasites. The complemented strain showed only a ~3% (±6.0) reduction compared to controls, demonstrating that the growth defect was predominantly due to the loss of DCT1 (Figs 4D, 3G and S5D).

We next tested whether the predicted spinster-like domain of PfDCT1 could functionally complement the loss of TgDCT1. To this end, we generated a complemented DCT1-mAID cell line (DCT1-mAID/ChimDCT1-Ty) by inserting at the *UPRT* locus a construct encoding the PfDCT1 spinster-like domain flanked by the N- and C-terminal regions of TgDCT1, C-terminally fused to a 3×Ty tag and driven by the *T. gondii Tubulin1* promoter (S5E, S5F Fig). Correct integration was confirmed by genomic PCR and WB (S5G, S5H Fig).

IFAs analyses revealed strong ChimDCT1-Ty expression during parasite division, with only weak signal in non-dividing parasites. Although its localization appeared more diffuse than that of cDCT1-Ty and partially resembled an endoplasmic reticulum pattern, clear co-staining with the developing daughter cell IMC was observed. Upon IAA treatment, the endogenous DCT1-mAID-3HA protein was efficiently depleted, while ChimDCT1-3 Ty remained stably expressed. After 30 hours of IAA exposure, IMC morphology was indistinguishable from that of untreated parasites, indicating full rescue of the mutant phenotype (Fig 4E). Plaque assays further confirmed the functional complementation by the chimeric protein: under IAA treatment, plaque size in the complemented line was reduced by only ~16.8% (±8.2) compared to the untreated DCT1-mAID condition (versus ~91.2% (±0.5) reduction in the non-complemented DCT1-mAID+IAA condition) (Figs 4F and S5I). Together, these results demonstrate that DCT1 is essential for IMC morphogenesis and completion of the parasite lytic cycle, and that both wild-type and chimeric DCT1 constructs can fully restore the defects associated with DCT1 depletion.

## DCT1 knockdown impairs parasite motility and invasion

To assess the role of DCT1 in the lytic cycle, intracellular replication was quantified by counting parasites per vacuole 30 hours post-infection in TIR1 and DCT1-mAID lines ±IAA. DCT1 depletion significantly reduced vacuoles with 8, 16, or

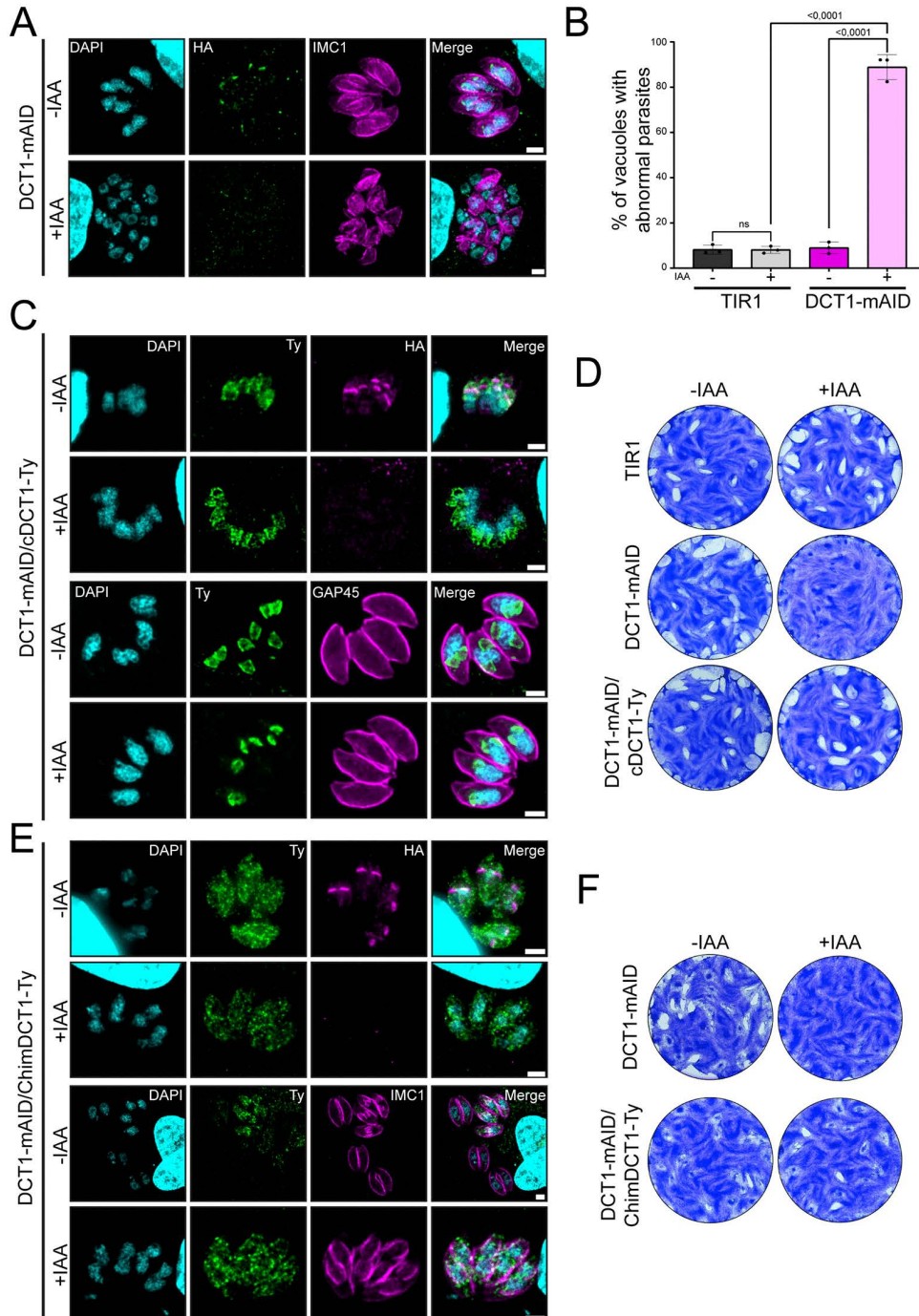

**Fig 4. Conditional DCT1 depletion is rescued by endogenous or chimeric DCT1 expression. (A)** IFA of intracellular DCT1-mAID parasites with or without IAA treatment, 30 hours post-infection, reveals DCT1 depletion and abnormal IMC morphology upon IAA exposure. DCT1 was labeled with anti-HA (green), the IMC with anti-GAP45 (magenta), and DNA with DAPI (cyan). Scale bar: 2 µm. **(B)** Graph displaying the percentage of vacuoles containing DCT1-mAID parasites with abnormal IMC morphology after 30 hours of IAA treatment. Statistical differences between groups were assessed using one-way ANOVA followed by Tukey's multiple comparisons (mean ± SD; n = 3 biologically independent experiments). **(C)** IFA of intracellular DCT1-mAID/cDCT1-Ty parasites ± IAA treatment, 30 hours post-infection. The complemented DCT1-Ty version remains stably expressed (Anti-Ty; green), while the endogenous DCT1-mAID-3HA signal (Anti-HA; magenta) is lost after IAA treatment, confirming efficient degradation. IMC morphology appears normal following 30 hours of IAA exposure, as shown by anti-GAP45 staining (magenta). DNA was stained with DAPI (cyan). Scale bar: 2 µm. **(D)**

Plaque assay comparing the lytic cycle efficiency of parental TIR1, DCT1-mAID, and the DCT1-mAID/cDCT1-Ty cell line ± IAA. Representative images from three independent experiments. **(E)** IFA of intracellular DCT1-mAID/ChimDCT1-Ty parasites with or without IAA treatment, 30 h after infection. The complemented ChimDCT1-Ty protein remains detectable (anti-Ty; green), whereas the endogenous DCT1-mAID-3HA signal (anti-HA; magenta) disappears upon IAA addition, indicating effective protein depletion. ChimDCT1-Ty displays a stronger signal in parasites undergoing division, and IMC architecture remains unaffected after 30 h of IAA exposure, as shown by anti-IMC1 staining (magenta). Nuclei were visualized with DAPI (cyan). Scale bar: 2 μm. **(F)** Plaque assay assessing the lytic-cycle capacity of the parental DCT1-mAID strain compared with the DCT1-mAID/ChimDCT1-Ty complemented strain in the presence or absence of IAA. Images shown are representative of three independent assays.

32 parasites. However, abnormal morphology in DCT1-mAID +IAA parasites hindered accurate counting, resulting in an increased number of vacuoles with 'unassigned' parasite counts (Fig 5A).

To determine whether DCT1-depleted parasites can initiate division, we quantified the proportion of vacuoles containing dividing parasites at 6, 12, 24, and 30 h post-infection. At early time points (6, 12, and 24 h), no significant differences were observed between IAA-treated and untreated DCT1-mAID parasites, indicating that DCT1 depletion does not impair division initiation. In contrast, at 30 h post-treatment, a higher proportion of vacuoles in IAA-treated DCT1-mAID parasites were undergoing division, suggesting a delay in division progression and accumulation of vacuoles in the dividing state (S6A Fig). To further test this hypothesis, we performed a shorter IAA treatment (12 h) to facilitate staging of daughter cell formation. Consistent with delayed progression, a higher proportion of parasites displayed late-stage daughter cell formation following IAA treatment compared with untreated DCT1-mAID parasites (S6B Fig).

Given the marked disorganization of vacuoles in DCT1-depleted parasites, we assessed the synchronicity of their division after 30h of treatment and found ~28% (±4.6) of DCT1-depleted vacuoles were asynchronous, compared to ~4% (±1.3) in controls (S6C Fig). Despite these defects, induced egress triggered by 5-Benzyl-3-isopropyl-1H-pyrazolo [4,3-d]pyrimidin-7(6H)-one (BIPPO) [89] was unaffected across strains and conditions (Fig 5B), but DCT1-depleted parasites failed to disseminate from lysed vacuoles, suggesting impaired gliding (Fig 5C). As microneme secretion is essential for both egress and gliding, we tested if DCT1 depletion affected this process [90]. Although a slight decrease can be observed, microneme secretion was not significantly changed across conditions in EtOH-stimulated assays (Figs 5D and S6D). To directly assess gliding, we evaluated motility after BIPPO stimulation. DCT1-mAID +IAA parasites displayed impaired gliding and a rounded morphology, rescued in the complemented line, confirming that the motility defect is due to DCT1 loss (Fig 5E). A similar phenotype was observed in Rapa-treated DCT1-Ty-U1 parasites (S6E Fig). Invasion assays showed a ~46% (±3.6) reduction in host cell invasion following DCT1 knockdown (Fig 5F), highlighting the key role of DCT1 in motility and invasion.

## DCT1 depletion disrupts parasite morphology by impairing IMC homeostasis and cytokinesis

To investigate the basis of motility and invasion defects, we examined IAA-treated DCT1-mAID parasites after egress and found abnormal morphologies, including rounded or double-headed forms (Fig 6A). Quantification showed that ~59% (±4.1) of IAA-treated DCT1-mAID parasites displayed abnormal morphology upon egress, compared to ~11% (±1.5) in untreated controls (Fig 6B). Additionally, mutant parasites treated with IAA had a smaller length (3.9±0.96 μm) (S6F Fig) and higher width (2.57±0.73 μm) (S6G Fig) compared to untreated mutant parasites (5.6±0.60 μm and 2.26±0.36 μm, respectively). To assess the morphological consequences of DCT1 depletion, we compared the ultrastructure of extracellular DCT1-mAID parasites treated or not with IAA using U-ExM on extracellular parasites, co-staining either the IMC (GAP45) or the PM (SAG1) together with acetylated tubulin. This revealed disruptions in both the IMC and PM, disorganized SPMTs, and, in some cases, multiple SPTMT networks within a single IMC or membrane compartment, indicative of a cytokinesis defect. Additionally, DCT1-depleted parasites showed disrupted connections between the IMC and the PM (Fig 6C). Ultrastructural analysis by TEM revealed IMC disruptions, including basal complex enlargement or lack of IMC signal at this site, together with a clear cytokinesis defect (Figs 6D and S6H). Collectively, our data show that the loss of DCT1 disrupts IMC morphology, likely by destabilizing both the IMC and basal complex, leading to defective cytokinesis.

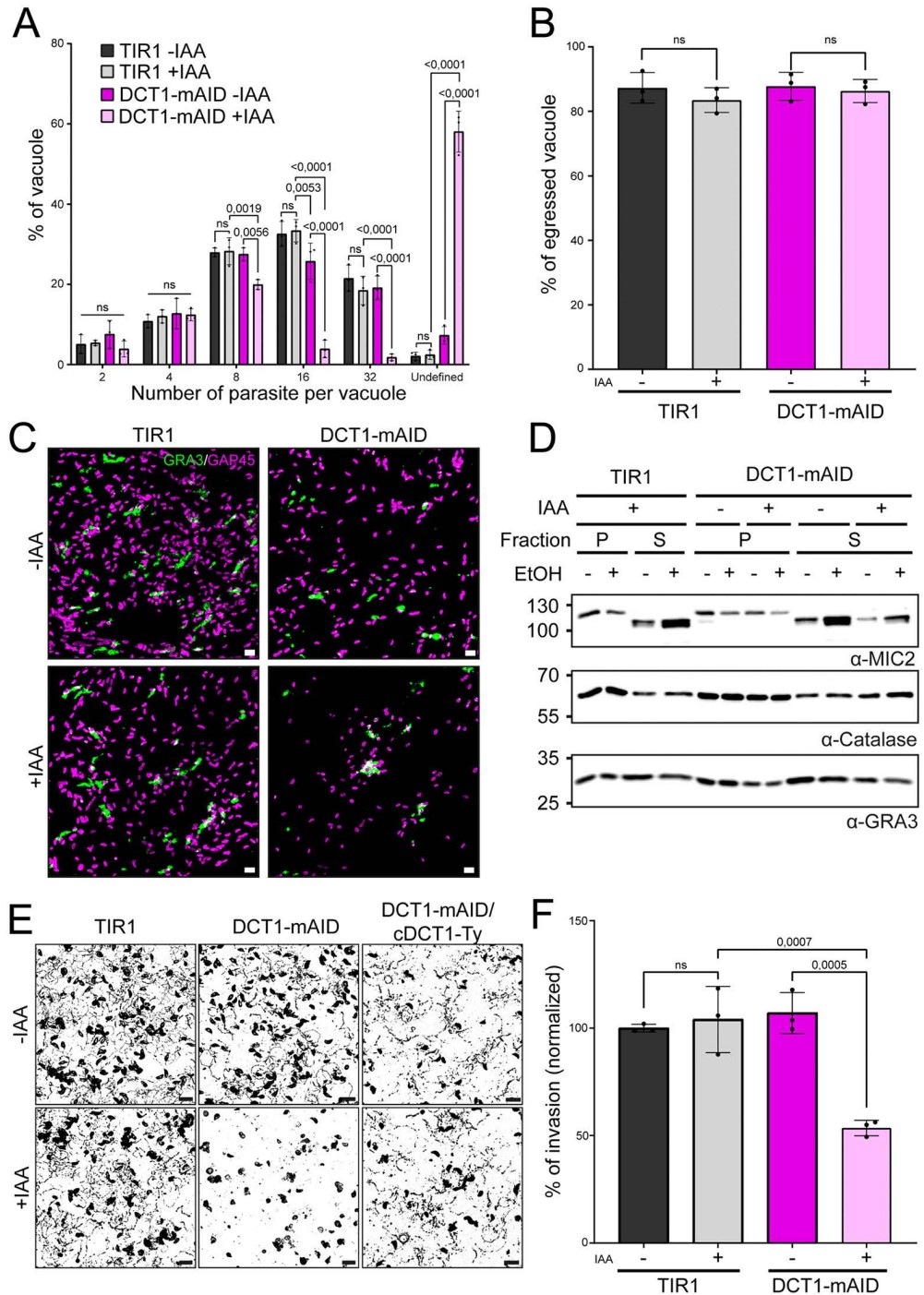

**Fig 5. DCT1 depletion leads to morphological defects and impairment in motility and invasion. (A)** Quantification of the number of parasites per vacuole at 30 hours post-invasion for parental TIR1 and DCT1-mAID strains±IAA. Vacuoles showing severe morphological defects, in which the number of parasites could not be determined, were classified as containing an undefined number of parasites. Statistical differences between strains for each category were assessed using two-way ANOVA followed by Tukey's multiple comparisons (mean±SD; n=3 biologically independent experiments). **(B)** Graph showing the percentage of ruptured vacuoles following treatment with the egress inducer BIPPO. Statistical differences between groups were analyzed using one-way ANOVA followed by Tukey's multiple comparisons (mean±SD; n=3 biologically independent experiments). **(C)** IFA of BIPPO-induced egress in host cell cultures infected with TIR1 or DCT1-mAID parasites±IAA for 30 hours. Parasites were stained with anti-GAP45 antibodies (magenta), and ruptured PVs were labeled with anti-GRA3 antibodies (green). Images are representative of three biologically independent replicates.

 

Scale bar: 10 µm. **(D)** Western blot analysis of microneme secretion from TIR1 and DCT1-mAID parasites ± IAA following ethanol stimulation (EtOH). MIC2 secretion was assessed by detecting MIC2 in the pellet (P) and supernatant (S) fractions using an anti-MIC2 antibody. Anti-GRA3 was used as a control for constitutive secretion, and Catalase as a cytosolic control. **(E)** Representative images of parasite motility following BIPPO treatment, comparing TIR1, DCT1-mAID, and DCT1-mAID/cDCT1-Ty ± IAA for 30 hours. Scale bar: 10 µm. **(F)** Percentage of intracellular ± IAA pretreated TIR1 and DCT1-mAID parasites 30 minutes post-infection. The percentage of invasion for each condition is normalized to the results obtained for the TIR1 -IAA condition. Statistical differences between groups were analyzed using one-way ANOVA followed by Tukey's multiple comparisons (mean ± SD; n = 3 biologically independent experiments).

## Disrupted IMC alters the morphology and distribution of endosymbiotic organelles

We first examined the biogenesis and positioning of secretory organelles in DCT1 depleted by staining for RON9 and ROP5, GRA3, and MIC2, as markers of the rhoptry neck and bulb, dense granules, and micronemes, respectively. While rhoptry biogenesis and localization appeared largely unaffected, parasites lacking IMC signal showed RON9 and ROP5 accumulation around the nucleus (Figs 7A and S7A). Since dense granule release depends on the apical annuli that are embedded in the IMC, we examined parasite secretion by staining for GRA3, which accumulates in the parasitophorous vacuole (PV) and intravacuolar network after release [25,91]. No significant changes in GRA3 staining were observed, indicating dense granule export still occurs despite IMC disruption (Fig 7B). Microneme biogenesis appeared not to be altered after DCT1 depletion and in parasites with a GAP45-positive IMC. However, in parasites lacking IMC signal, MIC2 staining adopted a dispersed, non-polarized, dot-like pattern (Fig 7C).

We then investigated whether IMC defects impair partitioning of the two endosymbiotic organelles by co-staining GAP45 with apicoplast (ATRX1) and mitochondrial (F1-ATPase) markers. Following DCT1 loss, some parasites showed defective apicoplast fission and mis-segregation (Fig 7D), while mitochondrial morphology and distribution were also altered (Fig 7E). These results underscore the importance of IMC homeostasis for proper endosymbiotic organelle partitioning, but minimal effect on secretory organelles.

## Loss of DCT1 alters the localization and distribution of IMC subcompartment markers

To further probe IMC morphology defects, we examined specific IMC structures using either existing antibodies or generated new cell lines within the DCT1-mAID mutant background by tagging proteins associated with distinct IMC compartments. Specifically, we tagged AC9, located in the apical cap of the alveolin network in both maternal and DCs [44], and GAP50, which is present in the cortical alveoli of both maternal and DCs [46]. We also tagged Centrin2, which localizes to several key IMC-associated structures, the preconoidal rings, centrioles, the apical annuli, and the basal complex [92]. To assess IMC sutures, we tagged ISC3, which localizes to the IMC alveoli, and TSC2, which is part of the alveolin network [21]. Additionally, to investigate whether the presence of IMC-associated micropore protein was perturbed by DCT1 depletion we tagged MPP1 [27] (S8A, S8F Fig). Correct integration of each tagged locus was confirmed by genomic PCR and WB (S8G–S8L Fig). After 30 hours of DCT1 depletion, none of the tagged proteins showed any significant reduction in abundance (S8M, S8R Fig).

We first examined pellicle organization in the absence of DCT1, focusing on IMC and PM markers. Double staining for SAG1 and GAP45 in DCT1-mAID parasites revealed vacuoles containing parasites with intact nuclei and PM but lacking GAP45 signal, consistent with cytokinesis defects seen by U-ExM (Figs 8A and S7B). Co-staining of SAG1 with either IMC1 or GAP50-Ty confirmed this phenotype, suggesting DCT1 depletion disrupts coordination between IMC and PM during division (Figs 8B and 8C). Next, GAP50-Ty was used as a marker for DC biogenesis, co-stained with GAP45 to specifically visualize the maternal cell IMC. GAP50-Ty staining revealed maternal alveoli disruption as observed with GAP45. However, despite these structural abnormalities, DC formation still occurred, even in the absence of a well-defined maternal IMC (Fig 8D).

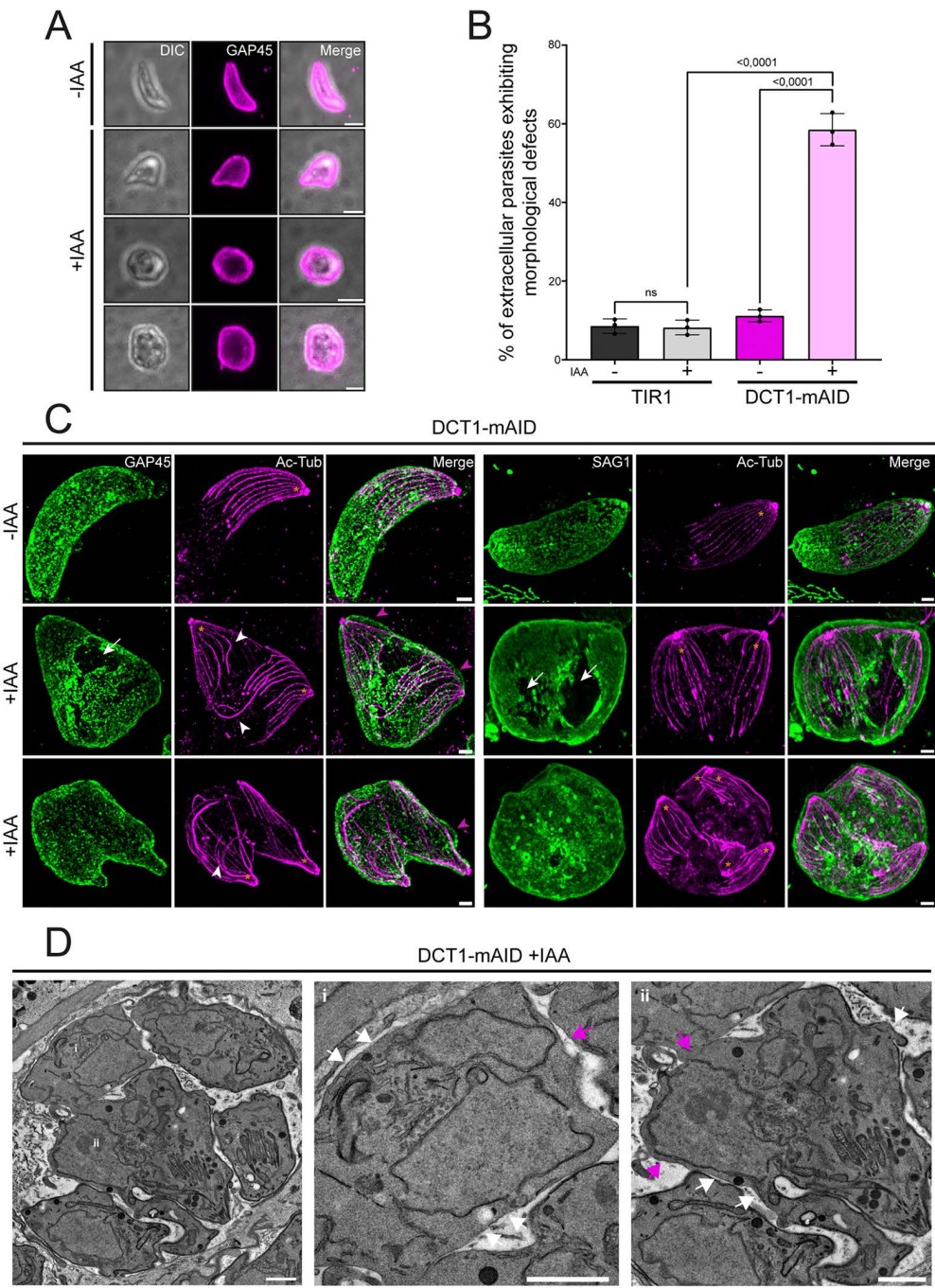

**Fig 6. DCT1 depletion impacts on IMC integrity and disrupts parasite cytokinesis. (A)** IFA showing the impacted morphology of DCT1-mAID extracellular parasites following IAA treatment. Anti-GAP45 antibodies were used to stain the parasite IMC (magenta). DIC: Differential interference contrast. Scale bars: 2 µm. **(B)** Graph showing the percentage of extracellular WT and mutant parasites displaying morphological defects. Statistical differences were evaluated using one-way ANOVA followed by Tukey's multiple comparisons (mean ± SD; n = 3 biologically independent experiments). **(C)** Ultra-Expansion microscopy showing morphological defects in extracellular DCT1-mAID +IAA parasites. Parasites were stained with antibodies against acetylated tubulin (Ac-Tub;magenta) and GAP45 (green), or with antibodies against SAG1 (green) under identical conditions. Orange asterisks indicate parasite tubulin networks. White arrows highlight IMC or PM disruptions. White arrowheads indicate SPMT distortions. Pink arrowheads show a lack of connections between the IMC and the SPMTs. Scale bars: 3 µm. **(D)** Electron microscopy of DCT1-mAID parasites upon IAA treatment shows rupture or absence of the cortical IMC, accompanied by IMC enlargement of the basal complex. White arrows indicate cortical IMC disruptions, magenta arrows mark enlarged or missing IMC signal at the basal complex. Scale bars: 1 µm.

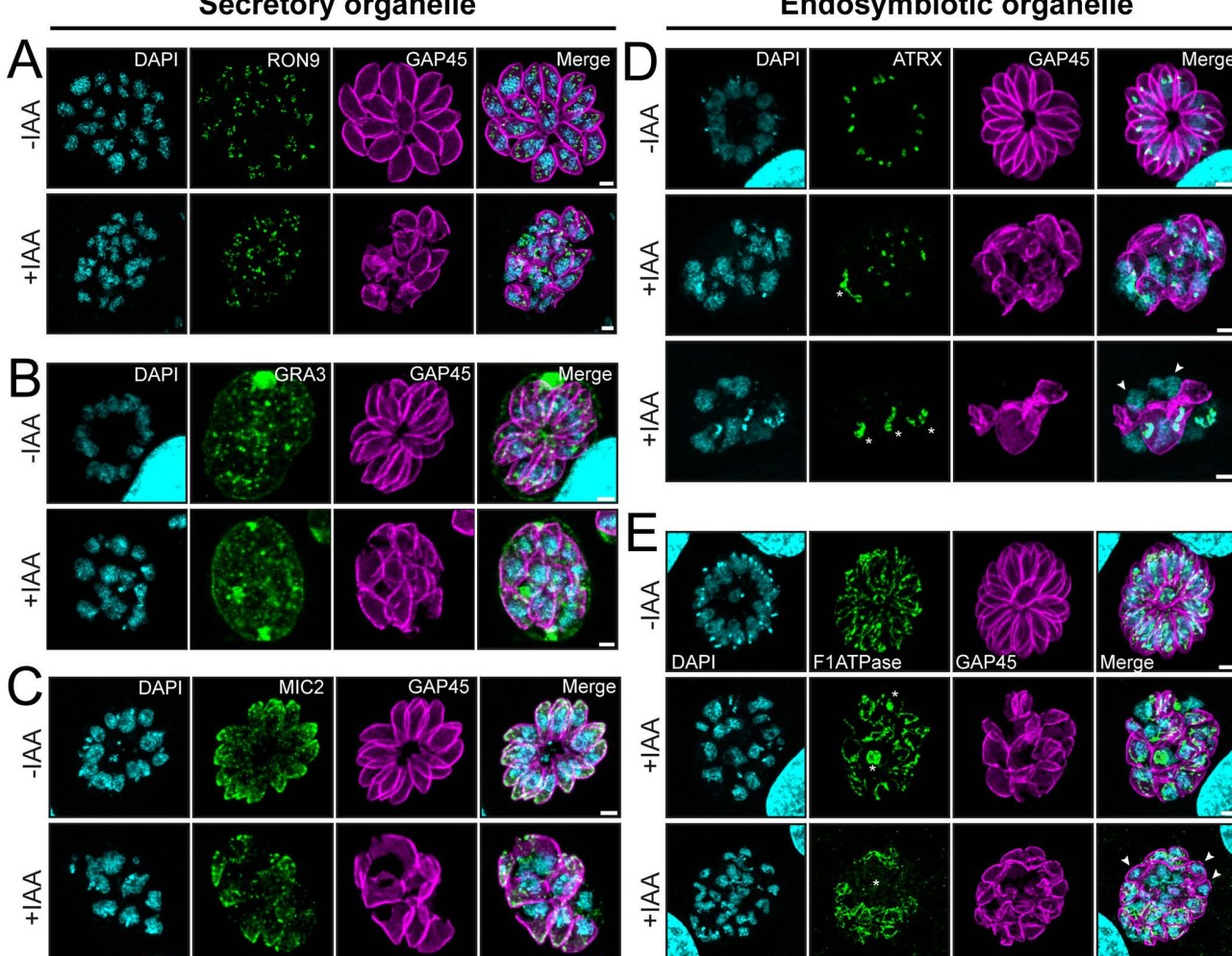

**Fig 7. DCT1 depletion alters the distribution and morphology of the apicoplast and mitochondrion.** IFAs comparing the biogenesis and local-ization of various organelles in DCT1-mAID upon treatment with IAA. DAPI marks the nucleus (cyan), GAP45 stains the IMC (magenta). **(A)** Rhoptry staining remains unaffected upon DCT1 depletion. Anti-RON9 has been used to stain the rhoptry neck (green). **(B)** PV and intravacuolar network dense granule staining remain unaffected in the absence of DCT1. Anti-GRA3 has been used to assess dense granule exocytosis (green). **(C)** Micronemal staining remains unaffected following DCT1 loss. Anti-MIC2 has been used to show the micronemes' localisation (green). **(D)** IMC morphogenesis defect upon DCT1 depletion impairs apicoplast repartition. Anti-ATRX (green) has been used to stain the apicoplast. Asterisks indicate severe defects in apicoplast morphology. Arrows highlight the parasite's nucleus devoid of apicoplast proximal staining. **(E)** IMC morphogenesis defect due to DCT1 loss impairs mitochondrial segregation and morphology. Anti-F1 ATPase stains the parasite mitochondria (green). Asterisks indicate mitochondrial morpholog-ical defects or partial disappearance of the mitochondrial signal. Arrows highlight parasites devoid of mitochondrial staining. Scale bars for all IFAs: 2 μm.

Since DCT1 appears early during DC formation, we next examined whether apical cap alveoli and alveolin network markers in DC were correctly positioned in the absence of DCT1. ISP1 staining showed partial loss of signal in the mother, while the DC signal remained intact (Fig 8E). In contrast, AC9-Ty, marking the alveolin network in both maternal and daughter apical caps, was unaffected (Fig 8F). These findings indicate that loss of DCT1 does not affect DC IMC biogenesis but primarily disrupts maternal IMC homeostasis, affecting the apical cap alveoli.

ISC3- and TSC2-tagged cell lines enabled the examination of IMC sutures in the context of DCT1 depletion; however, interpretation of the images proved challenging. Although ISC3 and TSC2 signals were still detectable, their organization

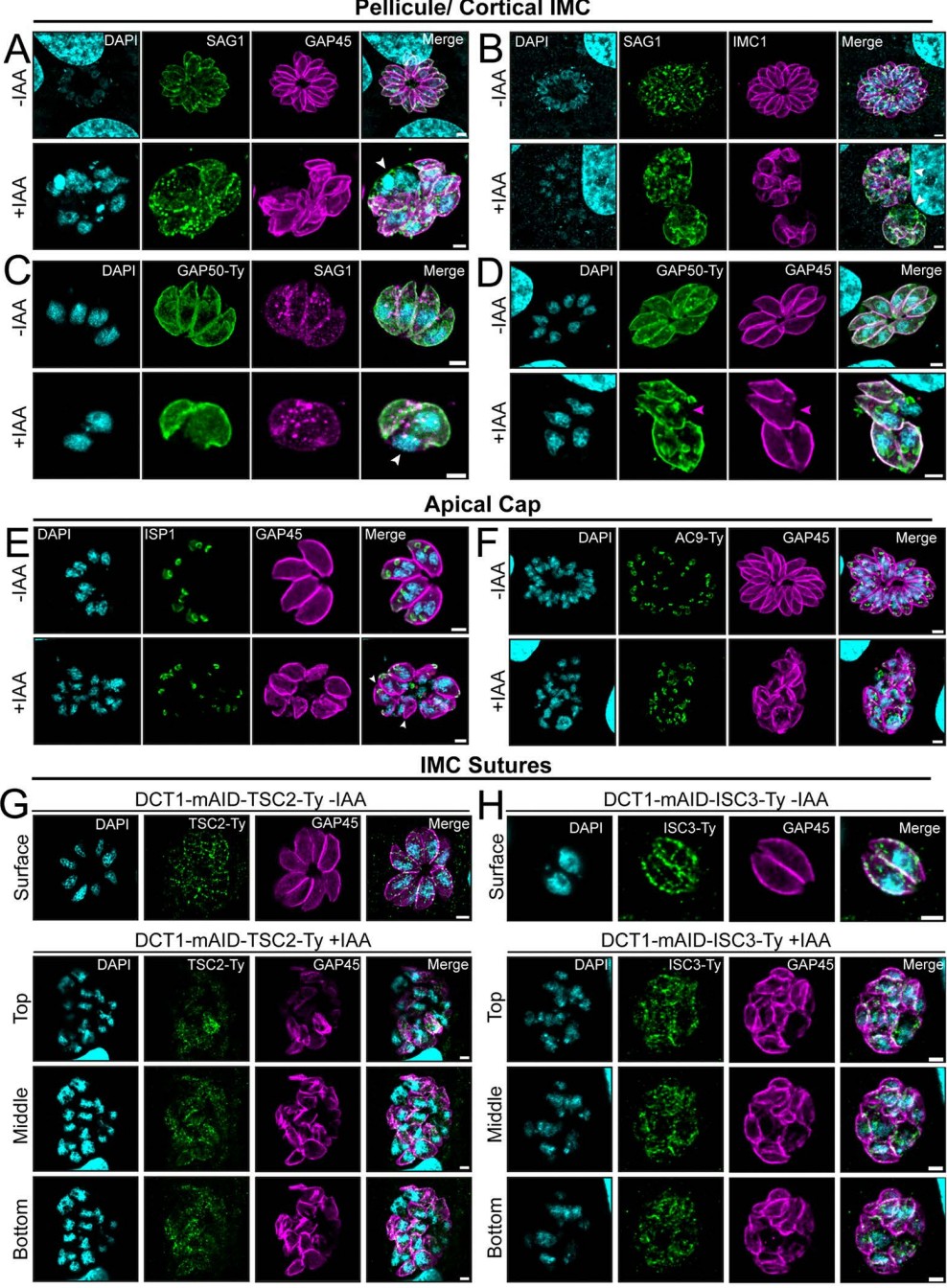

**Fig 8. DCT1 deletion disrupts the continuity of the pellicular space and affects IMC subcompartment homeostasis.** IFA images reveal partial disruption of the pellicle structure through co-staining of PM and IMC proteins. **(A)** GAP45 (magenta; anti-GAP45) and **(B)** IMC1 (magenta; anti-IMC1) were co-localized with the PM marker SAG1 (green; anti-SAG1). **(C)** GAP50-Ty (green; anti-Ty) was co-stained with SAG1 (magenta; anti-SAG1) in the DCT1-mAID-GAP50-Ty strain. White arrowheads indicate the presence of PM staining without IMC markers. **(D)** GAP50 tagging reveals disruption of the maternal cell IMC, whereas the daughter cell formation appears unaffected. GAP50-Ty (green; anti-Ty) was co-stained with GAP45 (magenta; anti-GAP45) in the DCT1-mAID-GAP50-Ty strain. Magenta arrowheads display ruptures of the maternal cell IMC. Red asterisks mark the forming daughter cell IMCs. **(E)** Depletion of DCT1 partially disrupts apical cap alveoli homeostasis. Anti-ISP1 stains the apical cap alveoli (green), and Anti-GAP45 stains the IMC (magenta). Arrowheads highlight mother cells lacking detectable ISP1 staining. Red asterisks uncover DC formation. **(F)** The apical cap alveolin network seems intact upon DCT1 deficiency. DCT1-mAID-AC9-Ty parasites labeled with anti-Ty to reveal AC9-Ty localization at the apical cap alveolin network (green) and Anti-GAP45 stains the IMC (magenta). Red asterisks display DC formation. **(G, H)** IMC sutures organisation is impaired following

DCT1 depletion. **(G)** TSC2-Ty and **(H)** ISC3-Ty were detected using anti-Ty antibody (green) in the DCT1-mAID-TSC2-Ty and DCT1-mAID-ISC3-Ty cell lines before or after IAA treatment. The IMC was stained using anti-GAP45 (magenta), and DNA was visualized using DAPI (cyan). Scale bars for IFAs: 2 μm.

into clearly defined sutures as observed in the DCT1 -IAA control was less apparent under DCT1 +IAA conditions (Fig 8G and 8H). MPP1-Ty stains the IMC-associated micropore structure, and still colocalized with the IMC signal, even in the presence of morphological abnormalities induced by DCT1 depletion (Fig 9A). Centrin2 localizes to the basal cup and apical annuli, became difficult to interpret upon DCT1 depletion. In parasites with intact IMC, Centrin2 localized to the preconoidal rings and near the nucleus (likely centrioles), but its apical annuli and basal complex signals were altered. In IMC-deficient parasites, Centrin2 appeared more diffuse and predominantly cytosolic (Fig 9B). To enhance visualization of the apical annuli and basal cup, U-ExM was employed. In untreated DCT1-mAID mutants, Centrin2 localized distinctly to each subcompartment. Upon DCT1 depletion, Centrin2 partially colocalized with GAP45, accompanied by an apparent size enlargement of the basal cup. Mislocalized annular structures were also observed, likely representing the basal cup, although the possibility of apical annuli staining cannot be excluded. Additionally, while apical annuli staining remained detectable, their architecture and spatial organization appeared disrupted (Fig 9C).

In summary, DCT1 depletion disrupts maternal alveolar homeostasis, leading to IMC disorganization and morphological abnormalities. As a result, several IMC-associated markers were impaired, absent, or mis-localized. In contrast, DC IMC neobiogenesis appears unaffected.

## Despite structural similarity to HsSPNS2, TgDCT1 appears to transport a distinct molecule

Next, we examined whether any known substrate-bound structures share similarity with DCT1. Using the AlphaFold-predicted model of DCT1, we found significant structural alignment with the cryo-EM structure of the *Homo sapiens* S1P transporter SPNS2 (HsSPNS2; PDB: 7YUB) [93] (Fig 9A). Notably, the predicted membrane-inserted region of DCT1 closely matches the experimentally resolved transmembrane domain of HsSPNS2 (Fig 9B). Similarly, the AlphaFold3 model of *P. falciparum* DCT1 (PfDCT1; PF3D7_0916000) (S2B Fig) overlaps strongly with HsSPNS2 (Fig 10C). These observations suggest that TgDCT1 and PfDCT1 share a high degree of structural similarity with HsSPNS2 and may perform analogous functions. The cryo-EM structure of HsSPNS2 bound to S1P reveals an acidic substrate-binding site, with a slightly less acidic region near the headgroup (Fig 10D). Electrostatic potential analysis of the substrate-binding cavities in TgDCT1 (Fig 10E) and PfDCT1 (Fig 10F) shows a similarly charged pocket capable of accommodating a lipid. Hydrophobicity mapping of these cavities further reveals a comparable distribution of hydrophobic regions (Fig 10G–10I), reinforcing the notion that the physicochemical environment of the binding site is conserved across these transporters. Nevertheless, despite this strong structural alignment, clear differences in pocket size and depth are observed, along with variations in key substrate-binding residues known to be critical for S1P transport in HsSPNS2 (Fig 10G) [70,93]. Although biochemical and functional studies are still required to define its precise substrate and function, DCT1 could likely pre-serve a conserved role in lipid transport within Apicomplexa, while these differences suggest that it transports a distinct substrate.

## Untargeted lipidomic analysis reveals DCT1-related alterations in parasite lipid balance

To gain further insight into the lipid species potentially transported by DCT1, we performed untargeted lipidomics. This approach allows detection of lipidome changes following deletion of enzymes involved in lipid synthesis, lipid transport-ers at the plasma membrane, or regulators of lipid homeostasis [94–97]. However, it remained unclear whether down-regulation of an internal putative lipid transporter, such as DCT1, would produce detectable global lipidomic changes. To test this, we analyzed the lipid profiles of the parental line (+IAA) and the DCT1-mAID mutant line (- or +IAA) using

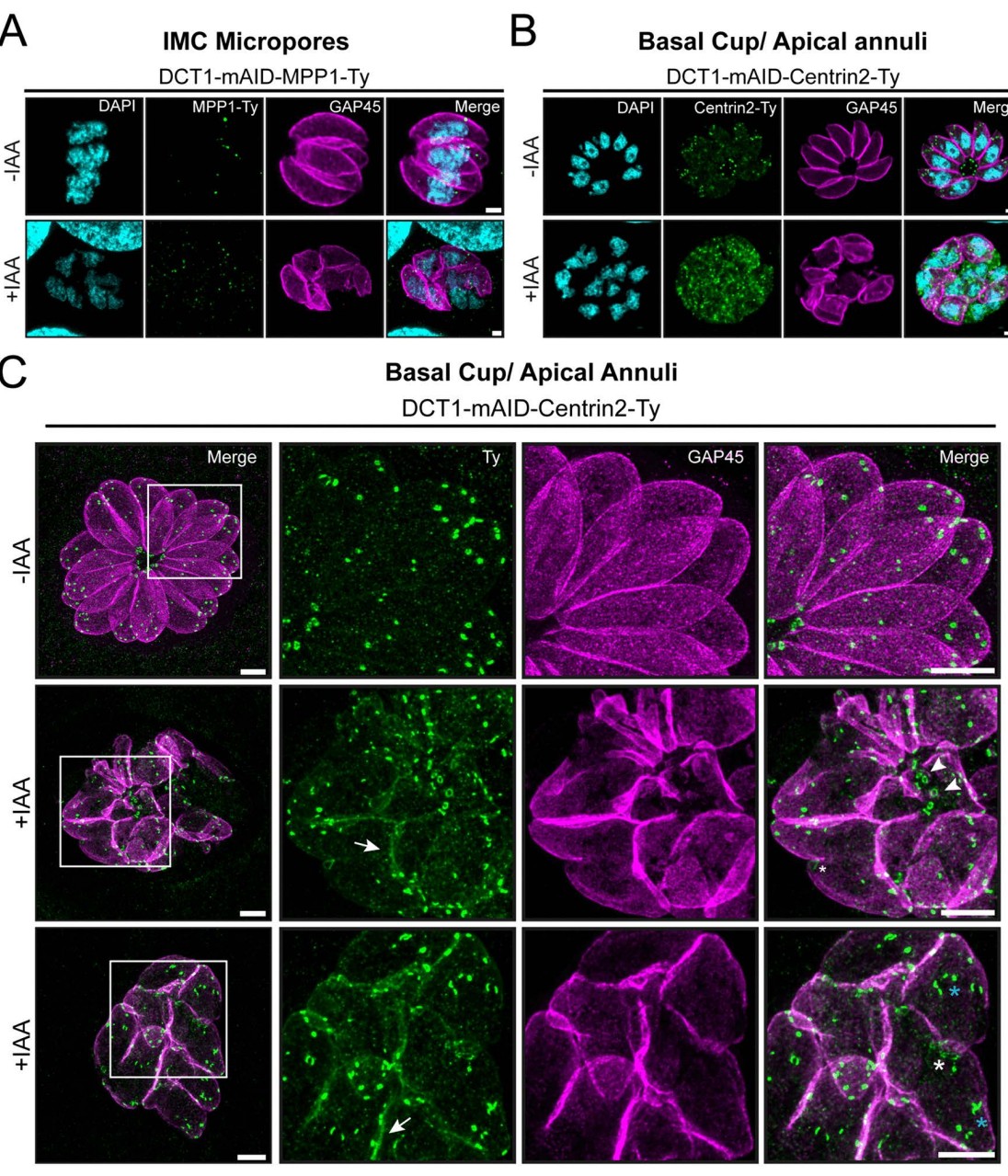

**Fig 9. DCT1 is required for the apical annuli organization and basal complex structure. (A)** The micropore protein MPP1 still colocalizes with the IMC after DCT1 knockdown. IFA showing MPP1-Ty (green) in the DCT1-mAID-MPP1-Ty cell line, detected with anti-Ty antibody. IMC and DNA were visualized with anti-GAP45 (magenta) and DAPI (cyan), respectively. **(B)** Centrin2 signal displays an altered pattern following DCT1 depletion. IFA of DCT1-mAID-Centrin2-Ty parasites stained with anti-Ty antibody (green), anti-GAP45 (magenta), and DAPI (cyan). Scale bars for IFAs: 2 μm. **(C)** U-ExM reveals defects in Centrin2 localization, along with abnormalities in the positioning of apical annuli and the positioning and morphology of the basal cup. DCT1-mAID-Centrin2-Ty parasites were stained with anti-Ty to reveal Centrin2-Ty (green). Insets disclose magnified regions from the corresponding images. White arrows indicate relocalization of Centrin2 at the cortical IMC. White arrowheads mark mispositioned structures resembling the basal cup. White asterisks highlight enlarged basal cups. Blue asterisks point out apical annuli with abnormal architecture. Scale bar: 10 μm.

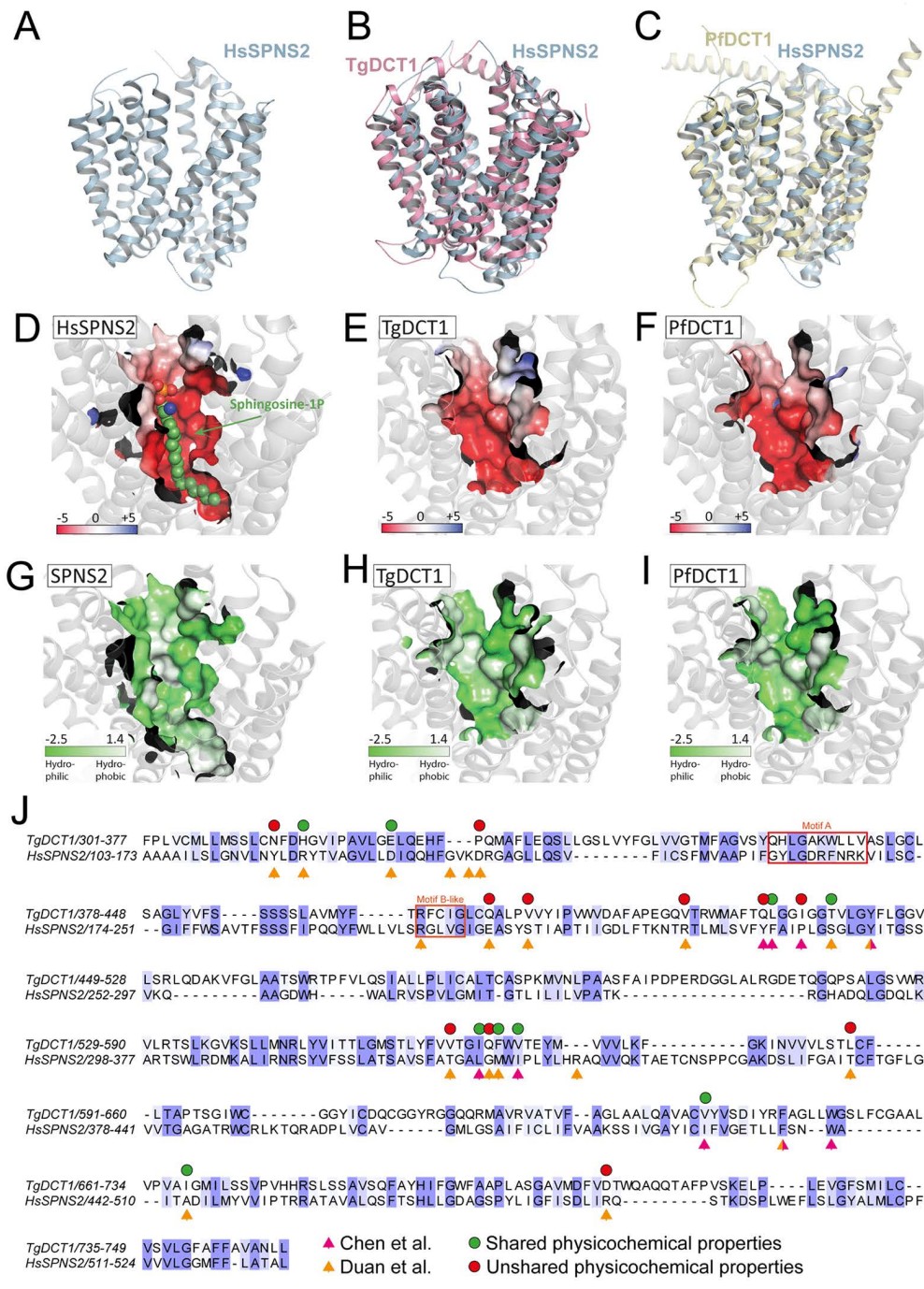

**Fig 10. TgDCT1, PfDCT1, and HsSPNS2 share overall structure and binding pocket electrostatic properties but likely accommodate different substrates. (A)** Structure visualization of HsSPNS2 (pdb: 7yub). **(B)** Overlay of TgDCT1 model with HsSPNS2. **(C)** Structure overlay of PfDCT1 model with HsSPNS2. **(D)** HsSPNS2 bound to S1P (green), with the binding pocket surface colored by electrostatic potential. **(E)** TgDCT1 model with pocket surface colored by electrostatic potential. **(F)** PfDCT1 model with pocket surface colored by electrostatic potential. Electrostatic potential was colored from −5 kT/e (red, negative) to +5 kT/e (blue, positive). **(G–I)** Binding pocket surfaces of HsSPNS2 **(G)**, TgDCT1 **(H)**, and PfDCT1 **(I)** colored by hydrophobicity (Eisenberg scale), from highly hydrophobic (white) to highly hydrophilic (green). **(J)** Sequence alignment of TgDCT1 and HsSPNS2 reveals conservation of the B-like motif, while the A motif and several key residues identified in HsSPNS2 as essential for S1P transport show poor conservation. Residue conservation (≥60% identity or similarity, BLOSUM62) is shown as a blue gradient, with darker shades indicating higher conservation. Magenta and orange arrows indicate key residues required for SPNS2 transport, as reported in the studies by Chen et al. and Duan et al., respectively. Green and red circles denote whether the physicochemical properties of the corresponding amino acids in TgDCT1 and SPNS2 are conserved or not in each case.

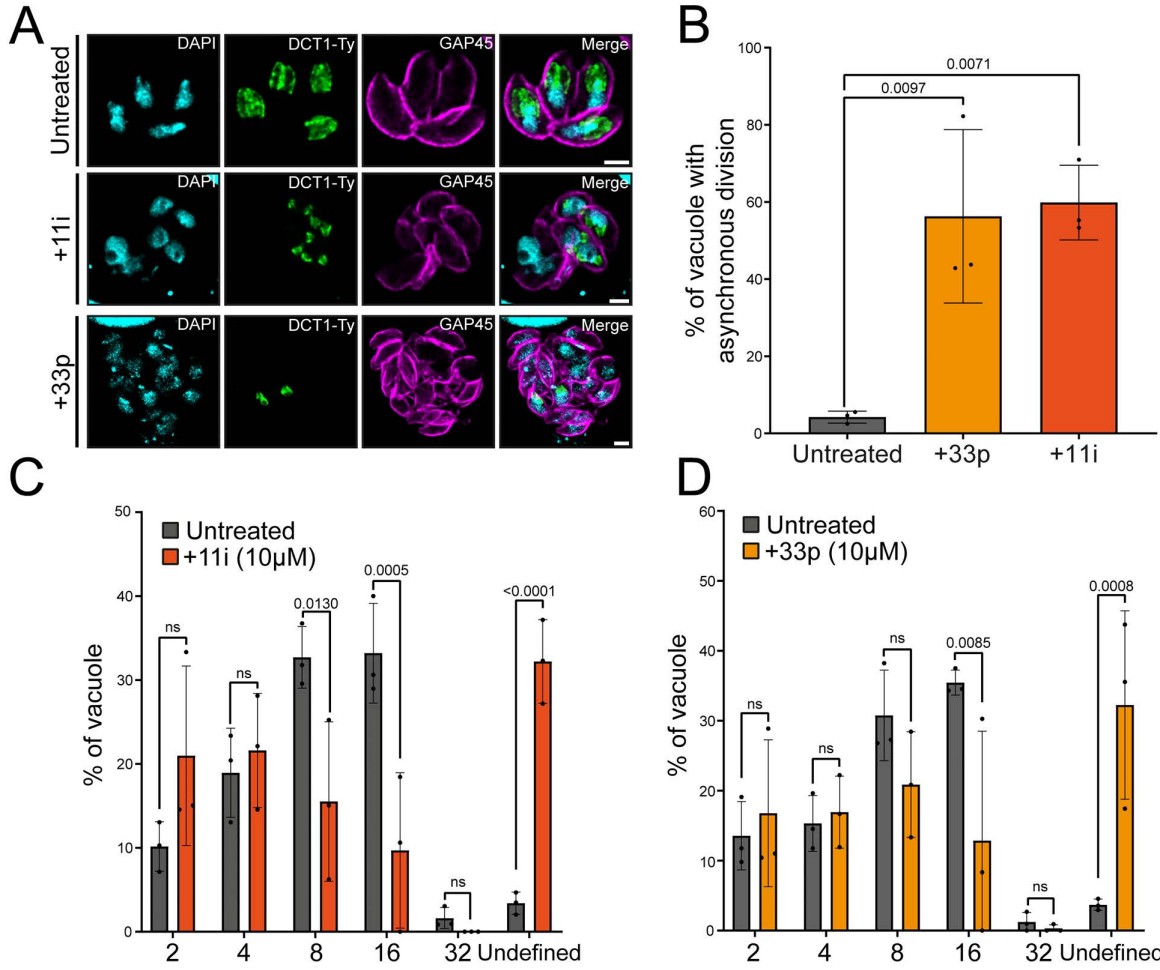

**Fig 11. 11i and 33p induce IMC defects and asynchronous division of *Toxoplasma gondii*. (A)** IFA images show that treatment with 11i or 33p disrupts IMC homeostasis and induces asynchronicity without affecting DCT1 positioning. Anti-Ty stains the DCT-Ty protein (green), and Anti-GAP45 stains the IMC (magenta). Scale bars for IFAs: 2 μm. **(B)** Quantification of asynchronously dividing vacuoles upon drugs treatement. Parasite division was assessed by IFA using an anti-IMC1 and anti-ISP1 antibody to visualize daughter cell budding. Statistical significance was determined using one-way ANOVA followed by Tukey's multiple comparisons test (mean±SD; n=3 biologically independent experiments). **(C, D)** Quantification of the number of parasites per vacuole at 24 hours post-invasion for TIR1 parasite treated with 11i (10μM) **(C)** or 33p (10μM) **(D)**. Vacuoles showing severe morphological defects, in which the number of parasites could not be determined, were classified as containing an undefined number of parasites. Statistical differences between treatments were assessed using two-way ANOVA followed by Tukey's multiple comparisons (mean±SD; n=3 biologically independent experiments).

reversed-phase liquid chromatography coupled to high-resolution mass spectrometry (LC-HRMS). Over 650 lipid species were annotated, and Orthogonal Partial Least Squares Discriminant Analysis (OPLS-DA) was used to compare TIR1+IAA parasites with DCT1-mAID parasites under - or +IAA conditions, as well as DCT1-mAID parasites in the -IAA versus +IAA state. The OPLS-DA models revealed distinct lipid shifts between TIR1+IAA and DCT1-mAID -IAA or +IAA conditions (S9A, S9B Fig), as well as between DCT1-mAID -IAA and +IAA parasites, indicating that DCT1 depletion induces notable changes in the parasite lipid profile (S9C Fig).

Univariate analyses showed that the levels of ether phosphatidylcholines (etherPC), ether phosphatidylethanolamines (etherPE), hexosylceramides (HexCer), hexosylceramide non-sphingoid (HexCerNs), lysophosphatidylethanolamines

(LPE), phosphatidylcholines (PC), phosphatidylethanolamines (PE), phosphatidylinositols (PI), phosphatidylserines (PS), and sphingomyelins (SM) were not significantly altered under any of the tested conditions.

Comparison between TIR1 +IAA and DCT1-mAID -IAA condition was characterized by the accumulation of bis(mono-acylglycero)phosphates (BMP), diacylglycerols (DG), dihydrosphingosines (DHSph), ether triacylglycerols (etherTG), oxidized triacylglycerols (OxTG), phosphatidylglycerols (PG), and triacylglycerols (TG), together with a decrease in carnitines (CAR), ester and ether lysophosphatidylcholines (LPC, etherLPC), and N-acylethanolamines (NAE). These changes indicate that even untreated DCT1-mAID mutants exhibit adaptive lipid remodeling.

Upon auxin treatment, DCT1-mAID parasites exhibited reduced levels of BMP and OxTG and increased levels of CAR, non-hydroxy-dihydrosphingosine (CerNDS), non-hydroxy-fatty acid sphingosine ceramide (CerNS), Coenzyme Q (CoQ), etherTG, and lysophospholipids (LPG, LPI, LPS) compared with untreated DCT1-mAID (−IAA) parasites. However, changes in BMP, CAR, CerNS, DG, OxTG, and PG were not significant when comparing TIR1 +IAA with DCT1-mAID +IAA, indicating that these alterations largely reflect pre-existing lipid differences in the DCT1-mAID background. By contrast, the increases in CerNDS and lysophospholipids (LPG, LPI, LPS) were significant in DCT1-mAID +IAA parasites, while no significant differences were observed between TIR1 +IAA and DCT1-mAID -IAA conditions. Overall, these observations indicate that DCT1 depletion induces subtle changes in the *T. gondii* lipidome. Whether these changes result directly from DCT1 transport activity or reflect downstream effects on overall lipid homeostasis remains unclear. Nevertheless, the data highlight a clear contribution of DCT1 to lipid balance, particularly in the regulation of CerNDS and lysophospholipid levels.

### SPNS2 inhibitors mimic DCT1 depletion, disrupting *T. gondii* replication, and division synchrony

We first assessed the cytotoxicity of the SPNS2 inhibitors 11i and 33p on human foreskin fibroblasts (HFFs) monolayers. At 10 μM, neither compound induced detectable cytotoxic effects, whereas both showed dose-dependent cytotoxicity at 25 μM and 50 μM, as indicated by increased Propidium iodide–positive nuclei (S10 Fig). Using the non-cytotoxic 10 μM concentration, we examined effects on *T. gondii* tachyzoites. IFA analysis revealed abnormal IMC morphology and asynchronous division, while DCT1-Ty localization remained unchanged (Fig 11A). Quantification of vacuoles showed a marked increase in asynchronous division after 24 h of treatment, with 60% (±9.7) and 56% (±22) of vacuoles affected by 11i and 33p, respectively, compared with 4% (±1.6) in untreated parasites (Fig 11B). Intracellular replication, assessed by counting parasites per vacuole at 24 h post-infection, showed that 11i reduced vacuoles containing 8 or 16 parasites, while 33p reduced vacuoles with 16 parasites (Fig 11C, 11D). Both treatments caused pronounced morphological abnormalities, resulting in a higher number of vacuoles categorized as "undefined," similar to observations in DCT1-depleted parasites.

To explore potential interactions with TgDCT1, we modeled 33p in the TgDCT1 structure based on its SPNS2 binding pocket. Both compounds had previously been modelled in complex with the human SPNS2 transporter [73,74], providing an initial structural framework. Building on this, the TgDCT1 model was first aligned with the SPNS2-33p complex and the localization of the small molecule was evaluated. According to the different poses of 33p in the SPNS2 binding pocket (S11A and S11B Fig), our visualization indicated that 33p could be accommodated within the modelled TgDCT1 structure (S11C and S11D Fig). Due to our inability to obtain a model of the SPNS2-11i complex, we employed a sequence conservation comparison approach using the amino acids of SPNS2 predicted to interact with 11i. The SPNS2 residues N112, Y210, S211, A214, and F234, F465 showed low sequence identity but shared some physicochemical properties with the corresponding aligned residues in TgDCT1 (S310, P407, V408, I411, T421, F686) (Fig 10J). This suggests that the observed effect may arise from a different mode of interaction with TgDCT1, or potentially from off-target effects on other parasite proteins. Together, these observations suggest that SPNS2 inhibitors 11i and 33p can interact with TgDCT1 and, at sub-cytotoxic concentrations, disrupt IMC morphology, impair intracellular replication, and induce asynchronous division without affecting DCT1 localization.

## Discussion

*T. gondii* possesses a repertoire of approximately 60 proteins containing an MFS domain, among which, around 15 have been predicted to be fitness-conferring based on genome-wide CRISPR screening data [83]. Here, we show that DCT1, a predicted fitness-conferring MFS transporter, localizes to the DC IMC and remains detectable during their development but rapidly disappears in mature parasites. In extracellular parasite, only a weak signal is observed near the nucleus, suggesting bulk protein degradation at the end of cytokinesis.

Two different inducible knockdown systems were used to deplete DCT1. The U1-mediated regulation allowed correct localization via Ty-tagging but resulted in slow protein depletion. In contrast, the mAID system enabled rapid protein depletion. However the protein is retained in the Golgi in the absence of IAA. Remarkably, this mis-localization did not produce any observable phenotypic defect. These results suggest that the C-terminal mAID-3HA tagging interferes with proper DCT1 processing and trafficking, and that a small amount of DCT1 at the DC IMC may be necessary and sufficient to maintain organelle homeostasis. Despite its limitations, the mAID system was instrumental in elucidating the topology of this polytopic protein and enabled rapid depletion.

In both inducible knockdown systems, DCT1 depletion leads to severe morphological abnormalities during parasite replication, including disorganized PVs and asynchronous division. While nuclear division is not significantly altered, the cytokinesis defect resulted in misshapen parasites. Importantly, the rhoptries, micronemes, and dense granules are still formed, and the apical positioning of these secretory organelles remains largely unaffected, which explains why processes such as microneme secretion and egress are not impaired. In contrast, gliding motility is severely affected. This defect is a consequence of altered parasite morphology following DCT1 depletion, characterized by reduced length, increased width, and the occurrence of multiheaded parasites, hallmarks of defective cytokinesis. Such a motility defect has been reported in *T. gondii* mutants with a cytokinetic abscission defect [98]. The profound impairment in gliding DCT1-depleted parasites is due to the dramatic alteration of parasite shape, reflected by SPMT distortions likely resulting from disrupted IMC–SPMT connectivity as well as disrupted IMC–PM connections, clearly documented by U-ExM and TEM. While SPMT organization supports gliding, it is not strictly essential [99]. However, the integrity of the IMC, where the glideosome is anchored, is crucial for motility.

Unlike what is observed for apical secretory organelles, recent studies established that dense granules are secreted at the apical annuli, which are closely associated with the IMC [22,25]. In DCT1-depleted parasites, GRA3 staining of the PV and intravacuolar network remains normal. This may indicate that secretion occurs either through still functional apical annuli or directly at the PM, which may become more accessible in the absence of an intact IMC.

In contrast to the aforementioned secretory organelles, the apicoplast and mitochondrion are not synthesized de novo but partitioned during DC formation, often closely associated with the developing IMC [40,48–50]. Following DCT1 knockdown, both organelles show division and morphology defects, likely secondary to impaired IMC integrity rather than direct DCT1 loss. While mitochondrial-IMC linkage has recently been described, the interaction between the apicoplast and the IMC remains unclear. The apicoplast may be anchored directly by the IMC or indirectly through the centrosomes, which orchestrate IMC biogenesis and coordinate organelle segregation. Notably, nuclei lacking IMC membranes were observed, suggesting that centrosome-IMC disconnection may disrupt nuclear positioning and apicoplast inheritance.

The final step in *T. gondii* division involves recycling of the maternal PM in conjunction with basal complex constriction, leading to cytokinetic abscission. However, DCT1-depleted parasites frequently displayed a PM signal and a nucleus but no detectable IMC signal across several markers, including IMC1, GAP50, and GAP45. Interestingly, a complete absence of IMC signal was observed at the basal pole in some parasites, a phenotype reminiscent of defects previously reported by EM in several basal complex mutant parasites [100,101]. This suggests that the basal complex may be unstable or misassembled, permitting continued growth without proper DC shaping as reported in BCC0 mutants [39]. Although not systematically assessed across all mutants, defects in components like MORN1, BCC4, DrpC, and Centrin2 have been linked to similar phenotypes, including multinucleation, cytokinesis failure, and abnormal morphology [39,102–106].

To investigate the role of DCT1 on the basal complex and the apical annuli, we endogenously tagged Centrin2 in the DCT1-mAID mutant. In the absence of DCT1, apical annuli appear disorganized, likely due to suture mis-organization or broader IMC defects, and Centrin2 is mis-localized along the cortical IMC. We also observed detached ring-like structures resembling mislocalized basal cups, as well as enlarged basal cups similar to those described in basal complex mutants or after cytochalasin D treatment [39,100,101]. These findings highlight a key contribution of DCT1 in basal complex stability. Given its daughter-specific localization and impact on morphogenesis, we initially hypothesized that it may disrupt daughter bud assembly. This hypothesis was based on phenotypes seen in early DC mutants, which show IMC disruption and cytokinesis defects [37,38,40–42]. However, marker analysis revealed no defects in the initiation of daughter IMC budding. Instead, we observed an accumulation of vacuoles containing dividing parasites after 30h of treatment, and a higher proportion of vacuoles with late-stage DC IMC development after 12h. These results suggest that DCT1 is not required for the onset of DC budding but is essential for the progression or completion of DC IMC assembly.

Importantly, we observed maternal IMC defects, notably partial loss of the apical cap marker ISP1, as observed in FBXO1, IMC29, and IMC32 mutants. Unlike in the IMC32 mutant, where severe IMC disorganization affects AC9 positioning, the absence of DCT1 does not alter the localization of this protein. This suggests that the daughter and maternal apical cap alveolin network is largely intact, whereas the maternal apical cap alveoli is perturbed. Based on these observations, DCT1 may transport a substrate that is not essential for early IMC budding but is critical for IMC composition and stability during the later stages of DC formation and once the mature parasite is formed. The progressive accumulation of IMC defects over successive division rounds suggests that even minor alterations in metabolite composition can compromise IMC integrity. While the first division in the absence of DCT1 produces largely normal alveoli, subsequent rounds begin to affect membrane homeostasis, as evidenced by the accumulation of late-stage dividing parasites after 12h despite initially normal IMC morphology. Over time, these changes in DC IMC composition may lead to deterioration of membrane properties, reflected by mother cell IMC disruption and altered lipid balance after 30h of infection, ultimately impairing both IMC morphogenesis and function.

Using protein BLAST searches and phylogenetic analysis based on the conserved MFS domain, we identified well-conserved DCT1 orthologs across Alveolata species. DCT1 forms a well-supported clade within coccidian parasites and is also clearly present in haemosporidian lineages, including *Plasmodium*, *Hepatocystis*, and *Haemoproteus*. Similarly, we identified orthologs in Cryptosporidiiae and Colpodellida, specifically in *Cryptosporidium parvum* and *Vitrella brassicaformis*, respectively. In contrast, sequences retrieved from Piroplasmids such as *Babesia* and *Theileria* did not cluster with bona fide DCT1 orthologs in the phylogenetic tree, suggesting they correspond to more distantly related or functionally divergent MFS transporters. Based on these phylogenetic results, we postulate that DCT1 originally had a structural or homeostatic role associated with the alveoli, later adapting to IMC-specific functions in Apicomplexa. Comparative studies of DCT1 orthologs would clarify whether its localization and function are broadly conserved.

AlphaFold modeling revealed strong structural similarity, not seen by BLAST, between TgDCT1 and HsSPNS2, a known S1P transporter. TgDCT1 also aligns closely with PfDCT1, supporting functional conservation. Binding pocket analysis suggests that apicomplexan DCT1 proteins would likely transport similar substrates. While key residues within the transporter cavity are conserved across alveolate orthologs, the critical SPNS2 residues required for S1P binding are not conserved in TgDCT1. These findings support the hypothesis that DCT1 functions as a transporter for a substrate likely distinct from S1P. Interestingly, the functional complementation with a second DCT1 WT copy or with a chimeric protein harboring the PfDCT1 Spinster-like domain restored the morphology and fitness defects observed upon DCT1 depletion, reinforcing the hypothesis that both TgDCT1 and PfDCT1 can transport the same substrate. Although the chimeric construct showed partial colocalization with the daughter cell IMC and displayed a pattern somewhat reminiscent of the endoplasmic reticulum, the protein was only detectable in dividing vacuoles. This likely indicates that the N-terminal region, the C-terminal region, or both sequences are required for proper regulation of protein expression, rather than a transcriptional regulatory phenomenon, since both the WT and chimeric complementation constructs were driven

by the tubulin promoter and only exhibited an expression pattern during cell division. In *Plasmodium*, the IMC is crucial at every stage of the life cycle [107,108]. Notably, several IMC-associated proteins have recently been characterized and shown to regulate parasite biogenesis in both sexual and asexual stages. Of relevance, PfDCT1 was identified at the IMC in a BioID proximity labeling study aimed at identifying PhIL1-interacting proteins [109] but has not been detected by the recent spatial proteome of *P. falciparum* schizonts [110]. Although Pf*DCT1* and its *Plasmodium berghei* orthologue (PBANKA_0817000, hereafter referred to as Pb*DCT1*) are not predicted to be essential genes based on *P. berghei* gene knockout screens [111] and random piggyBac transposon mutagenesis in *P. falciparum* [112], these proteins have not yet been fully characterized. In contrast, a functional screen of *P. berghei* transporters showed that PbDCT1 is dispensable for blood stages and male gametogenesis but essential for mosquito-to-mouse transmission, with severely reduced salivary gland sporozoites and no resulting infection [113]. The precise nature of the transported product and the consequences of its absence would thus clarify why PbDCT1 becomes crucial during *Plasmodium* transmission stages while being less important during asexual replication.

DCT1 possesses all the conserved features of a spinster-like transporter, including the predicted topology and the B-like motif within the MFS domain. Defining its substrate specificity and transport mechanism will be crucial to understanding how *T. gondii* supports its alveoli stability. While we attempted to express the protein in insect cells, these efforts were unsuccessful. This limitation, not uncommon when working with membrane transporters, underscores the technical challenges inherent to their functional characterization.

To explore potential alterations in the lipid composition of DCT1-depleted parasites, we performed untargeted lipidomics. This analysis is particularly challenging for an internal transporter like DCT1, which localizes specifically to the DC IMC and likely mediates uni- or bidirectional metabolite transfer between the cytoplasm and the nascent IMC. Indeed, if a substrate normally transported by DCT1 cannot cross this interface, it may be degraded, accumulated, or redistributed within the parasite, complicating the interpretation of observed lipidomic shifts. The lipidomic analysis reveals that the DCT1-mAID cell line exhibits constitutive lipid remodeling even in the absence of IAA, likely corresponding to an adaptive phenomenon due to a reduced amount of DCT1-mAID-3HA protein targeted to the building alveoli. DCT1 depletion further modifies the lipid profile, notably increasing CerNDS and lysophospholipids, while changes in other lipid species mostly reflect pre-existing perturbations in the mutant background. These results could indicate a secondary effect or a specific role for DCT1 in regulating sphingolipid and lysophospholipid levels, consistent with its essential function in DC IMC development and structural integrity.

Although *T. gondii* lacks a canonical SPNS2 ortholog and may not transport the same metabolite, lipid trafficking is likely critical for proper alveoli biogenesis in the parasite. Given that TgDCT1 and its alveolate orthologues retain conserved features of spinster-like transporters, known as potentially druggable targets, we evaluated the effects of two SPNS2 inhibitors, 11i and 33p, on *Toxoplasma*. Parasite treated with 11i and 33p exhibit IMC morphological defects resembling those of DCT1-depleted parasites, suggesting that *T. gondii* possesses mechanisms linked to IMC homeostasis that are sensitive to these SPNS2-targeting compounds. To our knowledge, DCT1 is the first putative MFS transporter reported to share strong structural homology with SPNS2, and the only MFS transporter whose depletion leads to such a dramatic IMC-homeostasis defect. We then modeled and visualized the potential interaction of 33p with the AlphaFold-generated TgDCT1 structure. The results suggest that the molecule has sufficient space to fit into the binding pocket of TgDCT1. In the case of 11i, the amino acids predicted to interact with SPNS2 are poorly conserved in TgDCT1 in terms of sequence identity, although some retain similar physicochemical properties. These results, based on protein modeling and potential interactions between the compounds and HsSPNS2, should be interpreted with caution, as they are purely in silico predictions. Direct binding assays are needed to confirm whether TgDCT1 is indeed the molecular target of these compounds. Nonetheless, these findings provide altogether compelling evidence that these inhibitors target TgDCT1-dependent transport and may also be effective against other alveolates of medical or environmental relevance that possess DCT1 orthologues. Engineering derivatives with enhanced specificity for TgDCT1 and reduced off-target effects in host cells could pave the way for the development of safer and more effective antiparasitic compounds.

In summary, our work identifies DCT1 as an essential regulator of IMC homeostasis. While its precise molecular function remains to be defined, DCT1 could potentially act as a lipid transporter, either by maintaining the correct composition and asymmetry of the IMC bilayer or by mediating lipid exchange at membrane contact sites between the IMC and other organelles such as the endoplasmic reticulum. Its evolutionary conservation across life-threatening apicomplexans and critical role in parasite survival highlight DCT1 as a central player in IMC biology and a promising target for therapeutic intervention.

## Materials and methods

### Parasite culture and transfection

*T. gondii* tachyzoites were cultured in HFFs (ATCC; CCD-1112Sk) with Dulbecco's Modified Eagle's Medium (DMEM) (Dominique Dutcher; 509052) supplemented with 5% of fetal bovine serum (Capricorn Scientific; FBS-16A), 2 mM glutamine (Thermo Fisher Scientific; 25030024), and 25 µg/mL gentamicin (Thermo Fisher Scientific; 15750045). The RHΔKu80 (RH) [114], RH TIR1 ΔKu80 ΔHXGPRT (TIR1) [88], and DiCre ΔKu80 ΔHXGPRT (DiCre) [87] were used to obtain all the transgenic strains generated in this study. *Escherichia coli* XL-10 Gold chemically competent bacteria (Agilent) were used for vector amplification. In this study, two strategies were employed to generate guide RNA (gRNA) constructs. The DCT1 and Centrin2 gRNA expression vectors were generated via PCR amplification using the Q5 Hot Start Site-Directed Mutagenesis Kit (NEB; E0554S) on the pSAG1::CAS9-GFP-U6::sgUPRT vector [115]. Other gRNAs used in this study were generated using the same vector with specific primer pairs (detailed in S2 Table), annealed and inserted into the vector at the BsaI restriction site. DNA constructs were amplified using designed primers (detailed in S2 Table) and KOD DNA polymerase (Sigma; KMM-101NV) to promote homologous recombination. These constructs contained ~30 bp of homology to the 3' coding sequence of the gene of interest, followed by the targeted insert, and ~30 bp of homology to the 3' UTR, thereby facilitating site-specific integration.

KI-DCT1-Ty strain: KI-DCT1-Ty strain was generated via transfection of the RH cell line using 30µg of pSAG1::CAS9-GFP-U6::sgDCT1 vector and purified KOD PCR amplicon using primers containing homology regions for the DCT1 gene (TGGT1_258700) on pLinker-2Ty-DHFR as a template.

DCT1-mAID strain: DCT1-mAID strain was generated via transfection of the TIR1 cell line with 30µg of pSAG1::CAS9-GFP-U6::sgDCT1 vector along with purified KOD PCR amplicon using primers containing homology regions for the DCT1 gene on pYFP-mAID-3HA [116] as template.

DCT1-Ty-U1 strain: DCT1-Ty-U1 strain was generated via transfection of the DiCre cell line with 30µg of pSAG1::CAS9-GFP-U6::sgDCT1 vector along with purified KOD PCR amplicon using primers containing homology regions for the DCT1 gene on pG152-KI-3Ty-lox-SAG1_3'UTR-HX [117] as template.

DCT1-mAID/cDCT1-Ty and DCT1-mAID/ChimDCT1-Ty strains: The cDNA corresponding to the WT version of the *DCT1* gene and the Chimeric *DCT1* gene were synthesized into the pTwist Amp High Copy vector (Twist Bioscience) using the EcoRI and NheI restriction sites. Both cDNAs were subsequently subcloned between the *EcoRI* and *NheI* sites into the UPRT-pTub1-G13-3Ty vector, which was derived from UPRT-pTub1-G13-Ty [118] by replacing the Ty epitope with a 3Ty tag. This placed these sequences under the control of the tubulin promoter, respectively generating the UPRT-pTub1-DCT1cDNA-3Ty and UPRT-pTub1-ChimDCT1-3Ty construct. For parasite transfection, DCT1-mAID parasites were electroporated with 60 µg of the linearized vector, along with 40 µg of a pSAG1::CAS9-GFP-U6::sgUPRT vector [115].

DCT1-mAID-AC9, Centrin2, ISC3, TSC2, and MPP1 tagging: To endogenously tag AC9, Centrin2, ISC3, TSC2, and MPP1 in the DCT1-mAID background using a 2Ty-DHFR cassette, gene-specific tagging constructs were amplified (primers listed in S2 Table) using the pLinker-2Ty-DHFR plasmid as a template. The DCT1-mAID strain was transfected with 30 µg of the different pSAG1::CAS9-GFP-U6::sgRNA-GOI plasmids encoding every gene-specific gRNA, together with purified KOD-generated PCR products.

To enrich the transfected populations, parasites carrying an HXGPRT cassette were selected using 25 mg/mL Mycophenolic acid (Sigma; M-5255) and 50 mg/mL Xanthine (Sigma; X-0626), while those carrying a DHFR cassette were

selected with 1 μg/mL pyrimethamine (Sigma; P-7771). DCT1-mAID/cDCT1-Ty parasites were selected using 5 μM 5-fluorodeoxyuridine (Sigma; F-0503). The clonality of the resulting parasite lines was verified through integration PCR using the couple described (S3 and S6 Figs and S2 Table), as well as IFAs. Proper expression of the tagged protein was confirmed by Western Blot analysis.

For all the assays involving the mAID system, the protein depletion was achieved by adding 500 μM of auxin (Sigma; I-2886). Depletion via the DiCre system was achieved by addition of 50 nM of Rapa (Calbiochem; 553210).

### Immunofluorescence assay

Intracellular tachyzoites cultured on coverslips with HFF monolayers were fixed for 10 minutes at room temperature using a fixation solution containing 4% paraformaldehyde (PFA) (Thermo Scientific; A11313.36) and 0.05% glutaraldehyde (Glu) (Sigma; G-5882). This was followed by a quenching step using 1X PBS with 0.1 M glycine (Biosolve; 7132391). The samples were then permeabilized with 0.2% TX-100 in 1X PBS for 20 minutes and blocked with 3% Bovine Serum Albumin (BSA) (AppliChem; A1391,0500) in 1X PBS for 20 minutes. Primary antibodies, diluted in 1X PBS/ 3% BSA (concentrations and origins listed in S2 Table), were incubated with the samples at room temperature under agitation. After three washes in 1X PBS, samples were incubated with secondary antibodies (origin listed in S2 Table), following the manufacturer's recommendations. Finally, the coverslips were washed three times for 5 minutes with 1X PBS containing 0.2% TX-100 (Sigma Aldrich; T9284-500ML) and mounted on microscope slides using DAPI Fluoromount G (SouthernBiotech; 0100-20)

### Western blot

HFF monolayers seeded in 6-cm Petri dishes were infected with freshly egressed parasites and cultured for 30 hours in the presence or absence of IAA or Rapa at the aforementioned concentrations. Following incubation, infected cells were scraped, pelleted at 1200 rpm for 10 minutes, and resuspended in sodium dodecyl sulfate-polyacrylamide gel electrophoresis (SDS–PAGE) buffer (50 mM Tris-HCl, pH 6.8, 10% glycerol, 2 mM EDTA, 2% SDS, 0.05% bromophenol blue, and 100 mM dithiothreitol). Samples were then subjected to SDS–PAGE and proteins were transferred onto nitrocellulose membranes for immunoblot analysis. Membranes were blocked with 1X PBS containing 0.1% (v/v) Tween-20 and 5% (w/v) nonfat dry milk, followed by incubation with primary and secondary antibodies diluted in the same blocking solution. Details of the concentration and origin of the antibodies used during the Western Blot are provided in S2 Table. Secondary antibodies conjugated with horseradish peroxidase were used following the manufacturer's instructions.

### Solubility assay

Several 6-cm dishes of heavily infected HFF monolayers with KI-DCT1-Ty or DCT1-mAID intracellular parasites were harvested by scraping 30 hours post-infection. The cells were pelleted, resuspended in 1X PBS, and divided into five aliquots. Each aliquot was pelleted again and resuspended in one of the following solutions: 1X PBS, 1X PBS/ 1 M NaCl, 1X PBS/ 0.1 M $Na_2CO_3$, 1X PBS/ 1% TX-100, or 1X PBS/ 1% SDS. The samples were subjected to five freeze-thaw cycles and incubated on ice for 30 minutes. The pellet and the soluble fraction were separated by centrifugation at $15,000 \times g$ for 30 minutes at 4 °C. Samples were resuspended in SDS–PAGE loading buffer containing 10 mM dithiothreitol. After protein separation by SDS–PAGE and transfer onto nitrocellulose membranes, the solubility of DCT1 was assessed using anti-Ty or anti-HA immunostaining. Controls included staining for soluble proteins (Anti-Actin), alveoli-anchored proteins (Anti-GAP45), and insoluble alveolin network proteins (Anti-IMC1).

### Plaque assay

Confluent HFF monolayers were infected with serial dilutions of freshly egressed *T. gondii* tachyzoites in the presence or absence of IAA or Rapa. Seven days post-infection, the infected monolayers were fixed with 4% PFA/ 0.05% Glu for

10 minutes, followed by neutralization with 1X PBS/ 0.1 M glycine. The fixed monolayers were stained with crystal violet (Sigma; C-0775) for 30 minutes and washed three times with 1X PBS. Images comparing the experimental conditions were captured using a Nikon camera. The lysis plaque areas were quantified by imaging the wells with an EVOS microscope (Thermo Fisher Scientific; AMF5000) and measuring their areas using ImageJ software (NIH, version 1.53c).

### Intracellular growth assay

Freshly egressed parasites were used to infect confluent HFF monolayers grown on glass coverslips, in the presence or absence of IAA. At 30 hours post-infection, cells were fixed with 4% PFA/ 0.05% Glu and processed for immunofluorescence using anti-GAP45 antibodies to visualize the parasites. For each condition, the number of parasites per vacuole was quantified by counting 100 vacuoles in three independent biological experiments.

### Microneme secretion assay

Several 6-cm dishes containing confluent HFF monolayers were heavily infected with freshly egressed parasites and cultured in the presence or absence of IAA for 30 hours. Infected cells were harvested by scraping and syringe-lysed using a 26G needle (Terumo; AN*2613R1). Parasites were pelleted and resuspended in DMEM media containing 2% EtOH or 2% Dimethyl sulfoxide and incubated at 37 °C for 10 minutes. Parasite pellets and supernatant (excreted-secreted antigen [ESA] fraction) were separated by centrifugation at 1000×g for 5 minutes. The ESA fraction was then separated and further centrifuged at 2000×g for 5 minutes to remove cell debris. Pellet fractions were washed once with 1X PBS to eliminate residual ESA. Both pellet and ESA fractions were analyzed by western blot using anti-MIC2, anti-catalase, and anti-GRA3 antibodies. The assay was performed in three independent biological replicates, and a representative result is presented in the manuscript. Images were acquired using ImageLab software (Bio-Rad), and microneme secretion was quantified by band densitometry using ImageJ software (NIH, version 1.53c).

### Egress assay

Freshly egressed parasites were allowed to infect confluent HFF monolayers and cultured for 30 hours in the presence or absence of IAA. BIPPO (10 μM) or Dimethyl sulfoxide was then added for 15 minutes. Following treatment, the medium was removed, and the coverslips were fixed with 4% PFA/ 0.05% Glu for 10 minutes, followed by neutralization with 1X PBS/ 0.1 M glycine. IFAs were performed using anti-GRA3 antibodies to label the PV and anti-GAP45 antibodies to stain the parasites. The average number of egressed vacuoles was determined by counting at least 100 vacuoles per condition in three independent biological experiments.

### Gliding assay

Syringe-lysed parasites, treated or not with IAA or Rapa for 30 hours, were pelleted by centrifugation at 1000 rpm for 10 minutes and resuspended in warm DMEM containing 10 μM BIPPO. Parasites were allowed to settle onto 0.1% gelatin-coated coverslips by centrifugation at 1000 rpm for 1 minute and incubated for 15 minutes at 37 °C. Following fixation with 4% PFA/ 0.05% Glu and quenching with 1X PBS/ 0.1 M glycine, immunodetection was performed using anti-SAG1 antibodies to visualize parasite trails.

### Invasion assay

Syringe-lysed parasites, pretreated or not with IAA for 30 hours, were allowed to settle by centrifugation at 1000 rpm for 1 minute and to invade confluent HFF monolayers for 30 minutes before fixation with 4% PFA/ 0.05% Glu for 7 minutes. Coverslips were quenched with 1X PBS/ 0.1 M glycine for 5 minutes. A first immunodetection was performed using anti-SAG1 antibodies on non-permeabilized cells to label extracellular parasites. Cells were subsequently fixed with 1%

formaldehyde in 1X PBS for 7 min, washed with 1X PBS, and permeabilized with 0.2% TX-100 in 1X PBS. After blocking for 20 min with 1X PBS containing 2% BSA, intracellular parasites were stained using anti-GAP45 antibodies. At least 100 parasites were counted per condition in each technical triplicate. The experiment was repeated in three independent biological replicates.

## Measurement of the extracellular parasite size

Freshly egressed parasites (± 48 hours of IAA treatment) were seeded on glass coverslips coated with 0.1% gelatin (Sigma; G-1890) and fixed with 4% PFA/ 0.05% Glu. Parasites were stained with anti-GAP45 antibodies in 1X PBS/ 0.2% TX-100/ 3% BSA. The length and width were determined for 100 parasites for each condition using ImageJ software (NIH; version 1.53c). The experiment was performed in three independent biological replicates.

## Transmission electron microscopy

HFF cells grown on round glass coverslip (13 mm in diameter) were infected with DiCre or DCT1-Ty-U1 parasites and allowed to grow in the presence or absence of Rapa for 30h before fixing with 2.5% glutaraldehyde and 2% paraformaldehyde for 1 h at room temperature. Once fixed, the samples were then washed with 0.1 M sodium cacodylate buffer (pH 7.4) five times for 5 min and postfixed with 1.5% potassium ferrocyanide and 1% osmium tetroxide in 0.1 M sodium cacodylate buffer (pH 7.4) for 1 h followed by 1% osmium tetroxide in 0.1 M sodium cacodylate buffer (pH 7.4) for another hour. Samples were then washed in double-distilled water twice for 5 min and en-*block* stained with aqueous 1% uranyl acetate for 1 h. Samples were then washed with ddH$_2$O twice and dehydrated in a graded ethanol series (2 × 50%, 70%, 90%, 95%, and 2 × absolute ethanol) for 10 min each. Cells were infiltrated with a graded series of Durcupan resin diluted with ethanol at 1:2, 1:1, and 2:1 for 30 min each and then twice with pure Durcupan for 30 min each. Cells were infiltrated with fresh Durcupan resin for an additional 2 h. A coverslip with cells was placed on a 1-mm high silicone ring filled with fresh resin, placed on a glass slide coated with mold-separating agent. It was then polymerized in an oven for 24 h at 65°C. Afterwards, the coverslip was removed from the resin disk by putting it in hot water (60°C) and then in liquid nitrogen. A laser microdissection microscope (Leica LMD) was used to outline the position of the PV on the exposed surface of the resin block. The area was then cut out from the disk and glued to a blank resin block with superglue. Using an Ultracut UCT ultramicrotome (Leica Microsystems) and a glass knife, the cutting face was trimmed. Finally, 70-nm ultrathin serial sections were cut with a diamond knife (DiATOME) and collected onto 2-mm single-slot copper grids (Electron Microscopy Sciences) coated with Formvar plastic support film. Sections were examined using a Tecnai 12 G$^2$ TEM (FEI, Netherlands) operating at an acceleration voltage of 80 kV and equipped with a side-mounted MegaView III CCD camera (Olympus Soft-Imaging Systems) controlled by iTEM acquisition software (Olympus Soft-Imaging Systems) at the Electron Microscopy Facility (PFMU) at the Faculty of Medicine at the University of Geneva.

## Ultrastructure-expansion microscopy

The U-ExM protocol applied to *T. gondii* tachyzoites was adapted from previously published methodologies [119]. Freshly egressed parasites deposited on poly-D-lysine-coated coverslips, or intracellular tachyzoites within infected HFF monolayers, were fixed using 1X PBS supplemented with 0.7% formaldehyde and 1% acrylamide for 3 hours at 37 °C. Polymerization of the gel matrix was carried out on ice using a monomer solution composed of 19% sodium acrylate, 10% acrylamide, 0.1% N, N′-methylenebisacrylamide, with the addition of 0.5% ammonium persulfate and 0.5% TEMED, following established protocols [119]. Following polymerization, the samples were incubated in denaturation buffer (200 mM SDS, 200 mM NaCl, 50 mM Tris, pH 9) at 95 °C for 90 minutes. The gels were then immersed in distilled water overnight to achieve expansion. The next day, the expansion factor was determined by measuring the gel diameter. Gels exhibiting a suitable expansion rate were incubated in 1X PBS to allow shrinkage for antibody labeling. Immunostaining was

performed using primary and secondary antibodies diluted in 2% BSA in 1X PBS, with incubations at 37 °C for 2 hours. Each antibody step was followed by three 10-minute washes in 1X PBS/0.1% Tween. After labeling, the gels were re-expanded in distilled water overnight before imaging. Images were acquired with a Leica TCS SP8 confocal microscope using a 100×/1.40 oil immersion objective (HC PL Apo). Z-stacks were collected and subsequently deconvolved using either Leica LAS X or Huygens software. ImageJ (NIH, version 1.53c) was used for post-processing, including the generation of maximum intensity projections.

### Image acquisition

Confocal IFA and U-ExM images were acquired using a Leica TCS SP8 STED microscope equipped with a 63×oil immersion objective (1.4 NA), at the Bioimaging Core Facility of the Faculty of Medicine, University of Geneva.

### Genome mining

The InterPro predicted MFS Spinster-like domain sequence of DCT1 was used as a query in BLASTp searches against the VEuPathDB database (https://veupathdb.org/) to identify putative orthologs in the following Alveolata organism: *Babesia bovis*, *Besnoitia besnoiti*, *Chromera velia*, *Cryptosporidium parvum*, *Cyclospora cayetanensis*, *Cystoisospora suis*, *Eimeria tenella*, *Gregarina niphandrodes*, *Haemoproteus tartakovskyi*, *Hammondia hammondi*, *Hepatocystis sp. ex Piliocolobus tephrosceles*, *Neospora caninum*, *Plasmodium falciparum*, *Porospora cf. gigantea A*, *Sarcocystis neurona*, *Theileria annulata*, *Vitrella brassicaformis.*

To expand the number of Alveolata species (not available on VeupathDB), BLAST searches were performed using BLASTp against the NCBI non-redundant protein database (https://blast.ncbi.nlm.nih.gov) on the following organisms: *Colponema edaphicum*, *Paramecium primaurelia*, *Perkinsus olseni*, *Polarella glacialis*, *Symbiodinium necroappetens,* and *Tetrahymena thermophila*. Gene IDs, scores, and e-values are shown in S1 Table.

### Taxonomy

The taxonomic nomenclature used was obtained from the NCBI Taxonomy database (https://www.ncbi.nlm.nih.gov/taxonomy), accessed in August 2025.

### Sequences alignment

Sequences alignment was performed using the command-line program ClustalW (http://www.clustal.org/clustal2/) [120]. The resulting alignments were imported and visualized in Jalview (version 2.11.4.1). Residues were colored either according to the BLOSUM62 similarity matrix with a conservation threshold of 60%, or based on the percentage identity between sequences.

### Phylogenetic analyses

Phylogenetic analyses were performed with the iqtree2 software [121] as follows:

MFS predicted sequences of the following organisms were aligned using the ClustalW method with default parameters for gap opening and gap extension: *Babesia bovis*, *Besnoitia besnoiti*, *Chromera velia*, *Cryptosporidium parvum*, *Cyclospora cayetanensis*, *Cystoisospora suis*, *Eimeria tenella*, *Gregarina niphandrodes*, *Haemoproteus tartakovskyi*, *Hammondia hammondi*, *Hepatocystis sp. ex Piliocolobus tephrosceles*, *Neospora caninum*, *Plasmodium falciparum*, *Sarcocystis neurona*, *Theileria annulata*, *Toxoplasma gondii*, *Vitrella brassicaformis.*

Only the MFS domain sequences were used to build the tree. Selection of the optimal model of sequence evolution for tree building was performed with ModelFinder among 1236 models. The tree was then built with the ultrafast bootstrap method of iqtree2 with 1000 replicates and was rendered using iTOL (https://itol.embl.de/) [122].

## Structural modeling and binding pocket analysis

TgDCT1 and PfDCT1 3D structures were predicted using AlphaFold3 [86] based on protein sequences retrieved from ToxoDB and PlasmoDB (https://plasmodb.org/plasmo/app). Models were ranked by pLDDT and TM-score, and only high-confidence predictions were used. Structural alignments were performed on PyMOL Molecular Graphics System, Version 3.0 Schrödinger, LLC. Putative binding pockets were identified using ChimeraX [123]. Electrostatic potential maps were calculated using the APBS Electrostatics Plugin [124] of PyMOL, with default parameters.

## Sample processing and LC-HRMS lipidomic workflow

**Sample preparation.** Confluent HFF cells grown in DMEM complete media supplemented with 5% FBS were infected with DCT1-mAID or TIR1 for 24 hours before the addition of IAA for a further 12 hours. Parasite and host metabolism were quenched through the addition of excess ice-cold PBS. All subsequent steps were carried out at 4°C or on ice. HFF monolayers containing intracellular parasites were scraped and parasites were released via multiple passages through a 26G needle. Parasite solutions were passed through a 3 µm exclusion size filter (Merck-Millipore, TSTP04700) to remove host cell debris and pelleted by centrifugation (2500g, 30 min, 4°C). Parasite numbers per sample were estimated using a Neubauer counting chamber. The pellet was washed 3x with cold PBS before being lysed with 500 µL isopropanol containing the internal standard LPC 18:1-d7 (1 µM, Avanti Polar Lipids, Alabaster, AL, USA) and stored at −80°°C until further processed. To obtain the lipid extract, the samples were vortexed for 1 min, shaken at in a Thermomix for 30 min at 10 °C and centrifuged for 15 min at 14'000 $g$ and 10 °C. The supernatants (450 µL) were transferred to new 1.5 Eppendorf tubes, evaporated to dryness in a SpeedVac (Thermo Savant SC210A, low temperature setting, 40 min) and stored at −80°°C until assayed. On the day of analysis, the dried extracts were resuspended in 120 µL methanol containing the internal standard Cer 18:1-d7/15:0 (1µM, Avanti Polar Lipids), shaken in a Thermomixer for 15 min at 1200 rpm and 10 °C and centrifuged for 15 min at 14'000 g and 10 °C. The supernatant was collected in LC-MS glass vials (TruView vials, Waters, Milford, MA, USA). Four technical replicates and three biological replicates per strain were used for analyses. Quality control (QC) samples were prepared by pooling 40 µL from each sample. From this pool, 10 QC samples were aliquoted to monitor acquisition stability throughout the sequence, 2 QC samples were used to condition the system before analyzing the biological samples, and 1 diluted QC sample (diluted 1:1 v/v with the reconstitution solvent) was included to identify potential contaminants

**LC-HRMS lipidomic measurements.** Untargeted lipidomic profiles were measured on a Vanquish Horizon LC system (Thermo Fisher, Waltham, MA, USA) with a BEH C18 Premier LC column (2.1 × 100 mm, Waters, Milford, MA, USA) connected to a high-resolution Exploris 120 Orbitrap mass spectrometer (Thermo Fisher). The mobile phase A was 10 mM ammonium acetate in acetonitrile/water 6:4 (*v/v*) and mobile phase B 10 mM ammonium acetate in isopropanol/acetonitrile 9:1 (*v/v*). The elution gradient ran with a flow rate of 0.4 mL/min from 20% to 60% B in 3 min, then to 85% in 7 min and to 97% in 5 min, followed by re-equilibration at the initial conditions for 4.5 column volumes. The LC flow was diverted to waste at 15 min. The column oven temperature was set at 50 °C and the injection volume was 2 µL. The mass spectrometer was operating in positive mode (3.4 kV), with internal mass calibration enabled at the start of each run. The source settings were 55 AU for the sheath gas flow, 13 AU and 350 °C for the auxiliary gas, 2 AU for the sweep gas, and the ion transfer tube temperature was set at 300 °C. S-lens RF was set at 70. Each sample was acquired in centroid mode with an alternating full scan at a 90,000 resolution on a mass range of 220–1200 *m/z* and top 4 DDA scans at 22,500 resolution with a stepped normalized collision energies of 10-30-60 eV. The AGC target was set at 100%, with an accumulation time of 80 ms for full scan and 60 ms for MS2 scans. MS2 scans were triggered with an intensity threshold of 20,000, an apex window of 35%, and a dynamic exclusion of 4 s. A custom exclusion list based on a blank sample was used to avoid fragmenting contaminants.

## Data pre-processing and curation

Data were pre-processed in MS-DIAL software v5.5.250221 [50]. Peak detection was performed with 0.01 and 0.025 Da mass tolerances for MS1 and MS2, a minimum peak height of 10000, a mass slice of 0.05 Da, and smoothing using a

linear weighted moving average with smoothing of 3 scans and minimum peak width of 5 scans. Deconvolution of MS2 data was performed with a sigma value of 0.5 and no abundance cutoff. Identification relied on an embedded lipid library with a mass tolerance of 0.01 and 0.025 Da (MS1 and MS2), and alignment was performed on the second QC sample with a retention time tolerance of 0.05 min and MS1 tolerance of 0.008 Da. MS2-matched lipid annotations were filtered for an accurate mass match < 5 ppm, redundant signals from adducts were removed, and expected elution patterns for lipids within the same class were verified. In addition, lipids with a missingness rate > 33% across the samples were excluded (missingness threshold = 5 × the signal in blank), as well as those presenting mean intensities in QCs < 1 × 10$^5$, CVs in QCs > 40%, and interquartile ranges = 0. A total of 659 lipids were retained in the final dataset. We applied PQN correction based on median QC intensities to account for variations in terms of biomass between the samples.

### Data exploration and statistical analysis

Lipidomic data exploration was done in SIMCA 17 (Umetrics Sartorius, Umea, Sweden). The data matrices were mean-centered and scaled to unit variance for principal component analysis for the acquisition quality control and OPLS-DA models (Raw and processed Data are detailed in S3 and S4 Tables). Graphs and statistical analyses were performed using GraphPad Prism 8.

### Assessment of in vitro drug susceptibility

Treatment with 11i (Med Chem Express; HY-162655), 33p (Med Chem Express; HY-150254), or Dimethyl sulfoxide was performed by mixing the compounds in DMEM supplemented with 5% FBS and applying them to infected or uninfected host cell monolayers, which were then incubated for 24 hours to assess cell susceptibility to the drugs. To assess HFF toxicity, following treatment with different concentrations of both drugs, the cells were washed once with DMEM/F-12 without phenol red (Thermo Fisher Scientific; 11039021). Propidium iodide (ThermoFisher Scientific; P3566) was then added to the same medium at a final concentration of 3 µg/mL, followed by a 5-minute incubation. The propidium iodide signal was subsequently observed using an EVOS microscope (Thermo Fisher Scientific; AMF5000) to visualize HFF nuclei and morphology.

### Supporting information

**S1 Fig. Conservation of the DCT1 gene across the superphylum Alveolata. (A)** Black circles indicate species with a conserved ortholog, defined using an arbitrary BLASTP *E*-value cut-off of 10$^{-20}$. In contrast, white circles indicate species in which no conserved ortholog was detected. Abbreviations: *Pf*, *Plasmodium falciparum*; *Hp*, *Hepatocystis piliocolobus tephrosceles*; *Ht*, *Haemoproteus tartakovskyi*; *Bbo*, *Babesia bovis*; *Ta*, *Theileria annulata*; *Bb*, *Besnoitia besnoiti*; *Cs*, *Cystoisospora suis*; *Hh*, *Hammondia hammondi*; *Nc*, *Neospora caninum*; *Sn*, *Sarcocystis neurona*; *Cc*, *Cyclospora cayetanensis*; *Et*, *Eimeria tenella*; *Cp*, *Cryptosporidium parvum*; *Gn*, *Gregarina niphandrodes*; *Pg*, *Porospora gigantea*; *Pp*, *Paramecium primaurelia*; *Tt*, *Tetrahymena thermophila*; *Cv*, *Chromera velia*; *Vb*, *Vitrella brassicaformis*; *Ce*, *Colponema edaphicum*; *Sne*, *Symbiodinium necroappetens*; *Pg*, *Polarella glacialis*; *Po*, *Perkinsus olseni*.
(TIF)

**S2 Fig. Sequences and structures conservation among DCT1 orthologs. (A)** Clustal alignment of the MFS domain from various DCT1 orthologues, highlighting differing levels of conservation of residues potentially involved in DCT1 substrate binding (magenta squares), with residues highlighted in shades of blue according to their percentage identity. **(B)** Predicted 3D structure of PfDCT1 (AlphaFold3), colored according to the pLDDT score. Magenta and cyan circles indicate the first N-terminal and the last C-terminal residues of the predicted coding sequence, respectively. **(C)** Overlay of the TgDCT1 (pink) and PfDCT1 (gold) MFS domains based on AlphaFold models.
(TIF)

**S3 Fig. Approaches to generate DCT1 mutants and PCR-based validation. (A)** Schematic representation of the homologous recombination strategy used to generate the KI-DCT1-Ty cell line. Primer sets used to detect the WT locus (1) and integration of the 2Ty-DHFR cassette (2) are indicated by black arrows. **(B)** Agarose gel showing PCR amplification of the WT or recombined DCT1 locus in RH and KI-DCT1-Ty strains. **(C)** Schematic of the homologous recombination strategy used to generate the DCT1-Ty-U1 cell line. Primer sets for detection of the WT locus (1), integration of the 3Ty-HX-U1 cassette (2), and confirmation of locus modification following Rapa treatment (3) are indicated by black arrows. **(D)** Agarose gel showing PCR products corresponding to the WT, recombined, and excised DCT1 locus after Rapa treatment in DCT1-Ty-U1 and DiCre strains. **(E)** Schematic representation of the strategy used to generate the DCT1-mAID cell line via homologous recombination. Primer sets used to detect the WT locus (1) and integration of the mAID-3HA-HX cassette (2) are shown by black arrows. **(F)** Agarose gel showing PCR amplification of the WT or recombined DCT1 locus in DCT1-mAID and TIR1 strains.
(TIF)

**S4 Fig. Analysis of DCT1 protein forms, inducible depletion, and localization using multiple tagging strategies. (A)** Representative images of WB on intracellular (In) and extracellular (Ex) parasite lysates from RH, KI-DCT1-Ty, DiCre, DCT1-Ty-U1, TIR1, and DCT1-mAID strains, showing the different forms of the DCT1 protein following different tagging strategies. DCT1-3Ty proteins from KI-DCT1-Ty and DCT1-Ty-U1 were detected using an anti-Ty antibody, while DCT1-mAID-3HA was detected using an anti-HA antibody. Anti-Catalase was used as a loading control. The black arrow indicates the full-length DCT1 protein, and purple arrows reveal additional forms of the tagged protein. **(B)** Rapid depletion of DCT1 in the DCT1-mAID strain following IAA treatment. DCT1-mAID-3HA was detected using an anti-HA antibody. Anti-Catalase was employed as a loading control. The black arrow indicates the full-length protein, and purple arrows highlight additional forms. **(C)** DCT1 (anti-Ty; green) does not colocalize with the maternal IMC (anti-GAP45; magenta) of KI-DCT1-Ty extracellular parasites. **(D)** In intracellular DCT1-mAID parasites (anti-HA; green), non-dividing parasites show no colocalization with the maternal IMC (anti-GAP45; magenta). **(E)** Upon overexposure, DCT1-mAID-3HA (anti-HA; green) is detected in the vicinity of the nucleus and at the DC IMC, while GAP45 (magenta) highlights the mother cell IMC. IFAs are stained with DAPI to visualize parasite nuclei. Scale bars: 2 µm. **(F)** Western blot analysis of DCT1-mAID-3HA solubility highlights the protein's presence at the Alveoli. Solubility assay has been done on intracellular parasites by using different buffers (PBS, 1 M NaCl, 0.1 M $Na_2CO_3$, 1% TX-100, and 1% SDS) in combination with total cell lysis by freeze-thaw cycles, resulting in a supernatant fraction (SN) and a pellet fraction (P). Anti-HA was used to reveal DCT1-mAID-3HA and Anti-IMC1 and anti-GAP45 antibodies were used as controls to assess the solubility of the alveolin network and alveoli-associated proteins, respectively. Anti-Actin was used as a control for soluble proteins.
(TIF)

**S5 Fig. Generation and functional characterization of DCT1 complementation cell lines. (A)** Schematic representation of the homologous recombination strategy used to generate the DCT1-mAID/cDCT1-Ty cell line. Primer sets used to detect the WT UPRT locus (1) and integration of the pTUB1-DCT1cDNA-3Ty cassette (2 and 3) are indicated by black arrows. **(B)** Agarose gel showing PCR amplification of the WT or recombined UPRT locus in DCT1-mAID and DCT1-mAID/cDCT1-Ty strains. **(C)** Immunoblot analysis of intracellular DCT1-mAID and DCT1-mAID/cDCT1-Ty complemented cell lines in the presence or absence of IAA for 30 h. Endogenous DCT1-mAID-3HA was detected using an anti-HA antibody, and the complemented version was detected using an anti-Ty antibody. Anti-catalase was used as a loading control. **(D)** Quantification of plaque assays performed on TIR1, DCT1-mAID, and DCT1-mAID/cDCT1-Ty cell lines in the presence or absence of IAA. Mean plaque area for each condition was normalized to the TIR1−IAA condition. Statistical analysis was performed using one-way ANOVA followed by Tukey's multiple-comparison test (mean±SD; n=3 biologically independent experiments). **(E)** Schematic representation of the predicted organization of the ChimDCT1 protein. Pink regions represent the N- and C-terminal portions of the TgDCT1 coding sequence, while the orange region corresponds to the MFS–Spinster-like domain of PfDCT1. Numbers indicate amino

acid positions along the full-length protein and mark the beginning and end of the PfDCT1 Spinster-like domain. **(F)** Schematic representation of the homologous recombination strategy used to generate the DCT1-mAID/ChimDCT1-Ty cell line. Primer sets used to detect the WT UPRT locus (1) and integration of the pTUB1-ChimDCT1-3Ty cassette (2 and 3) are indicated by black arrows. **(G)** Agarose gel showing PCR amplification of the WT or recombined UPRT locus in DCT1-mAID and DCT1-mAID/ChimDCT1-Ty strains. **(H)** Immunoblot analysis of intracellular DCT1-mAID and DCT1-mAID/ChimDCT1-Ty complemented cell lines in the presence or absence of IAA for 30 h. Endogenous DCT1-mAID-3HA was detected using an anti-HA antibody, and the complemented version was detected using an anti-Ty antibody. Anti-catalase was used as a loading control. **(I)** Quantification of plaque assays performed on DCT1-mAID and DCT1-mAID/ChimDCT1-Ty cell lines in the presence or absence of IAA. Mean plaque area for each condition was normalized to the DCT1-mAID -IAA condition. Statistical analysis was performed using one-way ANOVA followed by Tukey's multiple-comparison test (mean±SD; n=3 biologically independent experiments). (TIF)

**S6 Fig. DCT1 depletion impairs parasite division, motility, and morphology. (A)** Quantification of the number of vacuoles undergoing division at 6, 12, 24, and 30 h of treatment under DCT1-mAID−/+ conditions. A two-way ANOVA followed by Sidak's multiple comparison test was used to assess differences between groups (mean±SD; n=3 biologically independent experiments). **(B)** Distribution of dividing parasites at early, middle (Mid), or late stages of division after 12 h of treatment under DCT1-mAID−/+ conditions. Vacuoles containing parasites that were asynchronous or not distinguishable were categorized as undefined. A two-way ANOVA followed by Sidak's multiple comparison test was used to assess differences between groups (mean±SD; n=3 biologically independent experiments). **(C)** Quantification of asynchronously dividing vacuoles after 30 h of infection in IAA-untreated or IAA-treated TIR1 and DCT1-mAID conditions. Parasite division was assessed by IFA using an anti-IMC1 antibody to visualize daughter cell budding. Statistical significance was determined using one-way ANOVA followed by Tukey's multiple comparisons test (mean±SD; n=3 biologically independent experiments). **(D)** Percentage of microneme secretion in the supernatant fraction following ethanol stimulation. The microneme secretion for each condition is normalized to the results obtained for the TIR +IAA condition. One-way ANOVA followed by Tukey's multiple comparison was used to test differences between groups (mean±SD; n=3 biologically independent experiments). **(E)** Representative images showing parasite motility following BIPPO treatment, comparing DiCre and DCT1-Ty-U1 parasites with or without Rapa treatment for 30 hours. Scale bars: 2 µm. Quantification of **(F)** the length and **(G)** width of extracellular WT and DCT1-mAID parasites after 36 hours of treatment. One-way ANOVA followed by Tukey's multiple comparison was used to test differences between groups (mean±SD; n=3 biologically independent experiments). **(H)** Electron microscopy analysis of DCT1-Ty-U1 parasites treated with Rapa reveals rupture or absence of the cortical IMC, as well as enlargement and loss of IMC at the basal complex, accompanied by cytokinesis defects. White arrows indicate cortical IMC disruptions, magenta arrows mark enlarged or missing IMC signal at the basal complex, and orange asterisks highlight the presence of multiple apical complexes per parasite. Scale bar=1 µm. (TIF)

**S7 Fig. DCT1 depletion does not impair rhoptry biogenesis but partially disrupts pellicle organization. (A)** The rhoptry bulb was labeled using anti-ROP5 antibodies (green), while the IMC was stained with anti-GAP45 antibodies (magenta). Nuclei were stained with DAPI. **(B)** GAP45 (magenta; anti-GAP45) was co-localized with the PM marker SAG1 (green; anti-SAG1). Phase-contrast microscopy (Phase) was used to visualize the parasitophorous vacuole and the overall morphology of the parasites. White arrowheads indicate the presence of PM staining without IMC markers. Scale bars: 2µm. (TIF)

**S8 Fig. Generation and validation of DCT1-mAID strains co-expressing tagged IMC-associated proteins.** Schematic representations of the strategies used to obtain **(A)** DCT1-mAID-AC9-Ty, **(B)** DCT1-mAID-GAP50-Ty, **(C)** DCT1-mAID-Centrin2-Ty, **(D)** DCT1-mAID-ISC3-Ty, **(E)** DCT1-mAID-TSC2-Ty, **(F)** DCT1-mAID-MPP1-Ty. Primer sets used to

detect the WT locus (1) and integration of the 3Ty-DHFR cassette (2) are indicated by black arrows. Agarose gels showing PCR amplification of the WT or recombined targeted locus in DCT1-mAID and **(G)** DCT1-mAID-AC9-Ty, **(H)** DCT1-mAID-GAP50-Ty, **(I)** DCT1-mAID-Centrin2-Ty, **(J)** DCT1-mAID-ISC3-Ty, **(K)** DCT1-mAID-TSC2-Ty, **(L)** DCT1-mAID-MPP1-Ty. Immunoblot analysis of parental DCT1-mAID and **(M)** DCT1-mAID-AC9-Ty, **(N)** DCT1-mAID-GAP50-Ty, **(O)** DCT1-mAID-Centrin2-Ty, **(P)** DCT1-mAID-ISC3-Ty, **(Q)** DCT1-mAID-TSC2-Ty, **(R)** DCT1-mAID-MPP1-Ty cell lines in the presence or absence of IAA for 30 hours. Endogenous DCT1-mAID-3HA was detected using an anti-HA antibody, and the different tagged proteins in the DCT1-mAID background were detected using an anti-Ty antibody, with anti-Catalase used as a loading control.
(TIF)

**S9 Fig. Impact of DCT1 depletion on lipidomic signatures in *T. gondii*. (A-C)** Scatter plots of loadings from OPLS-DA models of **(A)** TIR+IAA and DCT1-mAID -IAA (1+3+0): R2X=0.73, Q2=0.96 **(B)** TIR+IAA and DCT1-mAID+IAA (1+2+0): R2X=0.65, Q2=0.99 **(C)** and DCT1-mAID -IAA and DCT1-mAID+IAA (1+2+0): R2X=0.64, Q2=0.98. The predictive component (pq(1)) reflects the variation directly associated with the experimental groups, while the orthogonal component (poso(1)) captures variation unrelated to group differences. Each point represents a lipid species colored by ontology, and its position on the horizontal axis indicates its contribution to the model. For example, in A, lipids with positive pq(1) scores are enriched in DCT1 -IAA, while those with negative scores are enriched in TIR+IAA. This visualization highlights the lipid classes most discriminating between the two conditions. **(D)** Graph depicting the difference in total ion count for each lipid species among the different conditions (TIR1+IAA, DCT1-mAID -IAA, and DCT1-mAID+IAA). These values were averaged, and TIR1 was set as 100%. Two-way ANOVA followed by Tukey's multiple comparison was used to test differences between groups (mean±SD; n=3 biologically independent experiments).
(TIF)

**S10 Fig. 11i and 33p treatment are not toxic at 10µM for 24h. (A)** Representative images of Propidium Iodide staining (red) of cells following 11i or 33p treatment at 10, 25 or 50 µM for 24h. Phase-contrast microscopy was used to assess HFF morphology. Scale bar: 100µm.
(TIF)

**S11 Fig. Visualization of the five predicted conformations of compound 33p within SPNS2 and TgDCT1. (A-B)** Superposition of the five predicted conformations of compound 33p within SPNS2 (PDB: 7YUB), shown as surface **(A)** and cartoon representations **(B)**. **(C-D)** Superposition of the five predicted conformations of compound 33p within the Alphafold3 TgDCT1 model, shown as surface **(C)** and cartoon representations **(D)**.
(TIF)

**S1 Table. Predicted *Toxoplasma gondii* MFSs, DCT1 orthologs, Alveolata MFS Fasta sequences.**
(XLSX)

**S2 Table. Primers and antibodies used in this study.**
(XLSX)

**S3 Table. RAW lipidomic data.**
(XLSX)

**S4 Table. Processed lipidomic data.**
(XLSX)

**S1 Data. Alveolata MFS Phylogenetic tree.**
(ZIP)

## Acknowledgments

We thank Dr. C. J. Tonkin (WEHI), Dr. P. Bradley (UCLA), Dr. J.-F. Dubremetz, Dr. V. Carruthers (UMMS), and Dr. K. Hu (ASU) for providing antibodies. We are grateful to Prof. Philip Thompson for providing BIPPO and to Jean-Baptiste Marc (UNIGE) for technical assistance. We sincerely thank Dr. Moritz Treeck (GIMM) for the DiCre RH ΔKu80 line; Dr. Vern Carruthers for the RH ΔKu80 strain; Dr. David Sibley (WashU Medicine) for the RH TIR1 ΔKu80 ΔHXGPRT strain, the pSAG1::CAS9-GFP-U6::sgUPRT vector, and the pYFP-mAID-3HA construct; Dr. Maryse Lebrun (LPHI) for the UPRT-pTub1-G13-Ty vector; and Dr. Markus Meissner (LMU) for the pG152-KI-3Ty-lox-SAG1_3'UTR-HX construct. We thank the Bioimaging Core Facility team: François Prodon, Olivier Brun, and Nicolas Liaudet, for technical assistance. We acknowledge Joachim Kloehn (UNIGE) and Albert Tell i Puig (UNIGE) for critical reading of parts of the manuscript and helpful comments on the figures, as well as other members of our laboratory for constructive discussions. We also acknowledge ToxoDB and VEuPathDB for providing open access to data for the scientific community.

## Author contributions

**Conceptualization:** Syrian G. Sanchez, Dominique Soldati-Favre.

**Formal analysis:** Syrian G. Sanchez, Nicolas Hulo, Dominique Soldati-Favre.

**Funding acquisition:** Dominique Soldati-Favre.

**Investigation:** Syrian G. Sanchez, Romuald Haase, David J. Dubois, Margaux Héritier, Rachel Humann, Nicolas Hulo, Bohumil Maco, Isabel Meister, Oscar Vadas.

**Methodology:** Syrian G. Sanchez, Dominique Soldati-Favre.

**Project administration:** Dominique Soldati-Favre.

**Resources:** Dominique Soldati-Favre.

**Supervision:** Dominique Soldati-Favre.

**Validation:** Syrian G. Sanchez.

**Visualization:** Syrian G. Sanchez.

**Writing – original draft:** Syrian G. Sanchez.

**Writing – review & editing:** Romuald Haase, David J. Dubois, Margaux Héritier, Rachel Humann, Nicolas Hulo, Bohumil Maco, Isabel Meister, Leonardo Scapozza, Oscar Vadas, Dominique Soldati-Favre.

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
