## [Decision Letter · Decision Letter 0]

14 Oct 2025

PPATHOGENS-D-25-02190

A Spinster-like Transporter at the Inner Membrane Complex is critical for Toxoplasma cytokinesis, motility and Invasion

PLOS Pathogens

Dear Dr. Soldati-Favre,

Thank you for submitting your manuscript to PLOS Pathogens. After careful consideration, we feel that it has merit but does not fully meet PLOS Pathogens's publication criteria as it currently stands. Therefore, we invite you to submit a revised version of the manuscript that addresses the points raised during the review process.

Please submit your revised manuscript within 30 days Dec 13 2025 11:59PM. If you will need more time than this to complete your revisions, please reply to this message or contact the journal office at plospathogens@plos.org. Please include the following items when submitting your revised manuscript:

We look forward to receiving your revised manuscript.

Kind regards,

Aoife T. Heaslip, Ph.D

Guest Editor

PLOS Pathogens

Tracey Lamb

Section Editor

PLOS Pathogens

Sumita Bhaduri-McIntosh

Editor-in-Chief

PLOS Pathogens

orcid.org/0000-0003-2946-9497

Michael Malim

Editor-in-Chief

PLOS Pathogens

orcid.org/0000-0002-7699-2064

**Additional Editor Comments:**

Overall, the reviewers agreed that this study would be of significant interest to the readers of PLOS pathogens, that the data was presented and described in a clear and concise manner and the experiments were performed to a high standard.

The reviewers agreed that there were several inconsistencies that should be addressed, namely: (1) The inconsistency between the localization of the Ty tagged and HA-mAID tagged versions of DCT1. This information has significant implications for future studies aimed at determining this proteins mechanism of action. (2) Direct evidence that DCT1 is a transporter. Authors suggest straight-forward experiments using inhibitors/probes to address this question. At a minimum, authors should temper their conclusions regarding this protein’s role as a lipid transporter.

**Journal Requirements:**

At this stage, the following Authors/Authors require contributions: Romuald Haase, David Dubois, Margaux Héritier, Nicolas Hulo, Bohumil Maco, Leonardo Scapozza, and Oscar Vadas. Please ensure that the full contributions of each author are acknowledged in the "Add/Edit/Remove Authors" section of our submission form.

4) We notice that your supplementary Figures are included in the manuscript file. Please remove them and upload them with the file type 'Supporting Information'. Please ensure that each Supporting Information file has a legend listed in the manuscript after the references list.

Potential Copyright Issues:

i) Figure 1B. Please confirm whether you drew the images / clip-art within the figure panels by hand. If you did not draw the images, please provide (a) a link to the source of the images or icons and their license / terms of use; or (b) written permission from the copyright holder to publish the images or icons under our CC BY 4.0 license. Alternatively, you may replace the images with open source alternatives. See these open source resources you may use to replace images / clip-art:

7) Please provide a completed 'Competing Interests' statement, including any COIs declared by your co-authors. If you have no competing interests to declare, please state "The authors have declared that no competing interests exist". Otherwise please declare all competing interests beginning with the statement "I have read the journal's policy and the authors of this manuscript have the following competing interests:"

**Reviewers' Comments:**

Reviewer's Responses to Questions

**Part I - Summary**

Reviewer #1: The paper by Sanchez et al, describes a new MFS-family transporter which they show is exclusively localised to daughter cells in Toxoplasma gondii. This conserved transporter appears similar to the human sphingosine-1-phosphate transporter. Using two conditional knockdown systems, they show in detail that this transporter is required for successful cytokinesis in these parasites and its loss leads to inhibition of the lytic life cycle. The phenotype, of localisation to the daughter cell but phenotype in the maternal cell, is unusual. The introduction and discussion are comprehensive and well written, although with relatively little information on the role of SPS. The figures are well presented and the results well described, however in many ways recapitulate the phenotypes seen in multiple other IMC mutants. One potential issue is the results are highly descriptive, with no significant attempt to identify a mechanism for this transporter. This is understandable, as functionalising transporters is difficult, however some efforts in this area would strongly support the conclusions of this manuscript. Overall however, the work is of excellent quality and appropriate for publication, and of interest to those researchers working in this area.

Reviewer #2: Members of the Major Facilitator Superfamily translocate metabolites. Toxoplasma gondii has MFS transporter genes that remain uncharacterized. One of these, the T. gondii Daughter Cell Transporter 1 (TgDCT1), is essential for daughter cell formation during parasite replication. This process involves construction of the IMC, a peripheral system of flattened vesicles that underlies the plasma membrane and is essential for parasite replication, motility, and host cell invasion. TgDCT1 localizes to daughter IMC and its conditional depletion disrupts the IMC, causing aberrant daughter cells morphology. Although depleted parasites are capable of host cell egress, they have severely impaired motility and host cell invasion. TgDCT1 harbors a predicted MFS spinster-like domain, is conserved across the Alveolata and shows structural similarity to the human sphingosine-1-phosphate transporter SPNS2, suggesting a role in lipid transport. This paper nicely shows that correct IMC biogenesis requires TgDCT1 and this influences cytoskeleton organization, daughter budding, pellicle domains and the basal complex. Given the extensive genetic tools that the authors have developed for this project, there are a few additional experiments I would like to suggest, along with a modification to the figure order.

Reviewer #3: This manuscript by Sanchez et al. focuses on TgDCT1, a member of the Major Facilitator Superfamily (MFS) of membrane transporters in Toxoplasma gondii. The authors demonstrate that TgDCT1 localizes to the inner membrane complex (IMC) in daughter cells during parasite division. To investigate the role of TgDCT1 in parasite growth, the authors generate an inducible knockout using the Cre-lox system. Additionally, they employ the auxin-inducible degron (AID) system to conditionally knock down TgDCT1, allowing rapid degradation of the tagged protein and facilitating phenotypic analyses. Their study shows that loss of TgDCT1 impairs parasite growth, motility, invasion, and endodyogeny. Knockdown parasites also exhibit altered morphology linked to changes in microtubule arrangement. The authors further use multiple cell cycle and organellar markers to characterize defects in TgDCT1-deficient parasites during cell division. Finally, phylogenetic analysis suggests that TgDCT1 is conserved across the alveolate clade and shares structural similarity with the human sphingosine-1-phosphate transporter.

Overall, this is a well-designed, thoroughly executed, and clearly written study. Below are my detailed comments and suggestions:

Figure 1:

• 1A: Please specify that the Western blot was performed using intracellular parasite samples.

• 1B: In the Results section, clarify why the authors conclude that TgDCT1 expression occurs earlier than ISP1 based on the data shown.

• General: No phase contrast or DIC images are included for immunofluorescence assays (IFAs) throughout the manuscript. Since visualization of the entire parasite morphology is important, the authors should include phase or DIC images alongside IFAs whenever possible.

• 1D: The Results should explain the significance of using different solvents to determine TgDCT1’s partitioning between pellet and supernatant fractions. Additionally, please explicitly state whether TgDCT1 behaves as a soluble or membrane-associated protein based on these results.

• 1D: A strong band at ~86 kDa is observed. Were these fractionation assays performed on extracellular or intracellular parasites? If extracellular, have the authors conducted IFAs on extracellular parasites to detect TgDCT1? Is there a signal observed under these conditions?

Figure 2:

• 2A and 2E: Please indicate in the figure or legend whether these assays were performed on intra- or extracellular parasites.

• 2B: Inclusion of phase or DIC images would greatly aid interpretation.

Figure 3:

• 3C: Including phase or DIC images would help demonstrate that parasites are lingering near the ruptured vacuole.

Figure 4:

• 4C: The legend or Results section should clarify the rationale for including panels of auxin-treated parasites.

• 4E: The authors need to better explain what they mean by "enlargement" of the basal complex.

Figure 5:

• Including a marker for rhoptry bulbs would strengthen the interpretation of Figure 5.

• Phase or DIC images would also be helpful here.

Additional Comments:

• Based on IFAs shown in Figure 7, the authors state that TgDCT1 depletion affects maternal alveolar homeostasis (line 326). However, since TgDCT1 predominantly localizes to daughter cells and not the mother cell, the authors should provide additional discussion explaining how the observed defects in maternal alveolar homeostasis might arise.

• Line 360: The authors do not provide direct experimental evidence that TgDCT1 functions in lipid transport. Therefore, they should significantly temper their conclusions regarding its role in lipid transport.

**Part II – Major Issues: Key Experiments Required for Acceptance**

Reviewer #1: The DCT1-mAID-HA line does not look like it is localised to the daughter IMC. If the localisation is important to the function of the protein (and the results here suggest that it is not?) then this huge misvocalisation would be expected to have a phenotype of some sort, and it is surprising that it does not. Can the authors confirm that it is degraded in the parental cells (as with the DCT1-Ty above) and that it is associated with the alveoli using the differential solubilisation?

The DCT1-mAID line efficiently degrades protein within 1 h, however from what I can see, all experiments were performed 24-30h post IAA addition. What are the first phenotypes seen upon DCT1 depletion? How is daughter cell budding affected at these early stages? Can the authors comment on

The strong structural homology of DCT1 to HsSPNS2 could allow the utilisation of small molecules to more deeply probe the function of this transporter. Have the authors considered using FTY720-P (e.g. PMID: 21084291) or one of the small molecule inhibitors (e.g. PMID: 39820269) to modulate the role of the potential transporter and then determining the phenotype?

There is a fairly long section based on the phylogeny and alphafold models. As these data do not inform experiments in this manuscript, this section could be reduced.

Reviewer #2: There are not major missing experiments that I would require for publication, but these proposed experiments leverage existing tools developed for this paper to test additional questions that arise from the data.

1. The proteolytic processing of TgDCT1 from 86 kDa to 60 and 40 kDa species detected by a C terminal epitope tag suggests that the N-terminus is removed to create the active ~60 kDa species. It would be valuable to use purified tagged protein species of each size to determine N-terminal sequences by MS.

2. The authors could use second copy knockdown system to probe the processing of TgDCT1 by creating truncation mutants. The transition in size to 60 kDa roughly corresponds to loss of the N-terminal fragment that lacks homology and the confidently predicted structure in alpha-fold. It could be interesting to follow the consequences of induced expression of a truncated copies of TgDCT1 in the mAID system to see if the N-terminal portion is required for correct localization to daughters and/or inhibition of activity. Also, given that Alphafold recognizes two types of domains in the transporter part of TgDCT1, I wonder if the second cleavage of 60 to 40 cuts these into separate parts.

3. Given the nice analysis of sites that are conserved and distinct in human HsSPNS2 (Figure 9), it would be worth analyzing the defined missense mutations identified in other spinster proteins (human, Drosophila) to find some that are in conserved positions to introduce mutations for complementation analysis in the mAID 2-copy system described in the MS.

4. With the caveat that the binding pocket is smaller in TgDCT1 (suggesting that it transports a distinct lipid), it would be an easy experiment to test whether treatment of Toxoplasma cultures with sphingosine 1-phosphate inhibitors (such as Fingolimod, an analog) could be tested to see if they enhance or phenocopy the loss of TgDCT1.

Reviewer #3: (No Response)

**Part III – Minor Issues: Editorial and Data Presentation Modifications**

Reviewer #1: Minor comments

The work is of a very high quality, and so I have few minor comments.

Line 202- It is possible that this is due to processing, or to change in the 3’ UTR which can affect protein abundance.

Line 447- The authors write: marker analysis revealed no defects in daughter budding upon DCT1 depletion however this result is not included in the text. Can this be included and quantified?

One potential way of determining the role of this transporter is to perform lipidomics at early timepoints, to look for changes. This would ideally be compared to a mutant with a similar phenotype which is not expected to change lipid composition. This however, is a potentially complex experiment, and so I offer it as a suggestion rather than a major comment.

Reviewer #2: 1. The authors describe using in silico methods to identify TgDCT1, but put the phylogenetic information, motifs and structure at the end where it seems like an afterthought. It would answer lots of questions that I had while reading the paper to move figure 8 to the front as figure 1. Figure 9 can stay where it is because it explores how TgDCT1 may function.

2. Given the strong likelihood that the non-conserved N-terminus is processed, it would be helpful to indicate in the Alphafold models and an alignment (Figure 8) where the “mature” protein would begin. Having this information in Figure 1 also helps the reader orient to the protein function. Also indicating the topology (C-terminus is cytoplasmic, allowing for the mAID system to work) in these models would be helpful.

Reviewer #3: (No Response)

PLOS authors have the option to publish the peer review history of their article (what does this mean? ). If published, this will include your full peer review and any attached files.

**Do you want your identity to be public for this peer review?** For information about this choice, including consent withdrawal, please see our Privacy Policy .

Reviewer #1: No

Reviewer #2: **Yes:** Naomi Morrissette

Reviewer #3: No

**Figure resubmission:**

**Reproducibility:**



---

## [Editor Report · Decision Letter 1]

9 Jan 2026

Dear Dr Soldati-Favre,

We are pleased to inform you that your manuscript 'A Spinster-like Transporter at the Inner Membrane Complex is critical for Toxoplasma gondii cytokinesis, motility and invasion' has been provisionally accepted for publication in PLOS Pathogens.

Best regards,

Aoife T. Heaslip, Ph.D

Guest Editor

PLOS Pathogens

Tracey Lamb

Section Editor

PLOS Pathogens

Sumita Bhaduri-McIntosh

Editor-in-Chief

PLOS Pathogens

orcid.org/0000-0003-2946-9497

Michael Malim

Editor-in-Chief

PLOS Pathogens

orcid.org/0000-0002-7699-2064

The authors of this paper have made substantial improvements from the first submission and have adequately address the reviewers concerns.

I have only one minor issue with the text. The authors created to parasite lines to study the function of DCT1. The first with a Ty tag and the second with a larger mAID tag. The localizations of these proteins are different. The first localizes to the daughter cell IMC, while the second localizes to the Golgi. On line 269 of the result section and line 546 of the discussion the authors state "Additionally, when the fluorescence signal was amplified, DCT1 could also be detected in developing daughter cells, indicating that the protein remains present at the growing alveoli during daughter cell formation (Fig. S4E)." Based on the images provided I am unconvinced that the AID tagged protein localizes to daughter parasites. Given that they successfully complemented this line (Fig. 4), it is reasonable to conclude that the phenotypes exhibited are due to loss of DCT1. It is my recommendation that the authors edit this statement to something more speculative, such as "Although DCT1-mAID was not detected in daughter cells, we speculate that a small amount of protein that is below our limit of detection is present in the IMC".
---

## [Editor Report · Acceptance letter]

Dear Dr Soldati-Favre,

We are delighted to inform you that your manuscript, "A Spinster-like Transporter at the Inner Membrane Complex is critical for Toxoplasma gondii cytokinesis, motility and invasion," has been formally accepted for publication in PLOS Pathogens.

Best regards,

Sumita Bhaduri-McIntosh

Editor-in-Chief

PLOS Pathogens

orcid.org/0000-0003-2946-9497

Michael Malim

Editor-in-Chief

PLOS Pathogens

orcid.org/0000-0002-7699-2064